# Pathophysiological Mechanisms of Diabetes-Induced Macrovascular and Microvascular Complications: The Role of Oxidative Stress

**DOI:** 10.3390/medsci13030087

**Published:** 2025-07-02

**Authors:** Bipradas Roy

**Affiliations:** Department of Physiology, Wayne State University, Detroit, MI 48202, USA; biroy@med.wayne.edu; Tel.: +1-615-290-1425

**Keywords:** diabetes mellitus, oxidative stress, reactive oxygen species, hyperglycemia, macrovascular disease, microvascular disease, atherosclerosis, diabetic retinopathy, diabetic kidney disease, endothelial dysfunction

## Abstract

Diabetic vascular diseases have emerged as a significant concern in medical research due to their considerable impact on human health. The challenge lies in the insufficient understanding of the intricate pathophysiological mechanisms associated with different forms of diabetic vascular diseases, which hampers our ability to identify effective treatment targets. Addressing this knowledge gap is essential for developing successful interventions. Unraveling the molecular pathways through which diabetes leads to microvascular and macrovascular complications in vital organs such as the heart, brain, kidneys, retina, and extremities is crucial. Notably, oxidative stress resulting from hyperglycemia is the key factor in initiating these complications. This review aims to elucidate the specific molecular mechanisms by which oxidative stress drives microvascular and macrovascular diseases and to highlight promising therapeutic advancements that offer hope for effective treatment solutions.

## 1. Introduction

Diabetes mellitus (DM) is a chronic endocrine and metabolic condition characterized by either insulin deficiency, insulin resistance, or a combination of both [1]. This disorder leads to elevated blood sugar levels, known as hyperglycemia, and can result in various diabetic vascular diseases over time. DM and its complications are rising at an alarming rate and have become a global threat to human lives, which requires urgent attention. In 2019, the global prevalence of diabetes was estimated at 9.3%, affecting 463 million people. This figure is projected to increase to 10.2% (578 million) by 2030 and 10.9% (700 million) by 2045 [2]. The increased prevalence of diabetes is mainly due to several factors, including an aging population, socio-economic growth, unhealthy dietary habits, and a sedentary lifestyle [3]. More than 90% of diabetes cases represent type 2 DM (T2DM) [4]. Diabetic complications affect multiple organ systems, including the heart, brain, kidneys, eyes, and peripheral blood vessels [5]. Diabetes poses a significant health burden primarily because of its vascular complications. These complications can be categorized into two groups (1) macrovascular complications, including coronary artery disease (CAD), cerebrovascular disease, and peripheral artery disease, and (2) microvascular complications, including diabetic kidney disease (DKD), diabetic retinopathy (DR), diabetic neuropathy, and cardiomyopathy [6].

The coexistence of multiple vascular complications results in a considerably worse prognosis [7]. Recent research into the complexities of diabetic complications has significantly enhanced our understanding of the disease’s underlying mechanisms. Yet, the growing fragmentation of medical specialties often leads to a narrow focus on isolated problems, risking a comprehensive view of the issue. It is crucial to adopt a holistic approach that investigates the interconnected nature of diabetic complications across various systems and vascular conditions. By doing so, we can better address the challenges posed by diabetes and improve outcomes for millions affected by this devastating disease.

Oxidative stress plays a significant role in diabetes-induced pathophysiology of vascular complications. In recent years, this condition has received considerable attention due to its profound implications for human health, especially its connection to diabetes.

Elevated reactive oxygen species (ROS) are the hallmark of oxidative stress in DM. At physiological concentrations, ROS serve as important second messengers that transmit intracellular signals involved in various biological processes. However, when ROS production becomes excessive and exceeds the capacity of antioxidant defense systems or when antioxidant enzymes are impaired, oxidative stress occurs [8,9]. ROS, which includes free radicals and other reactive molecules, are natural byproducts of normal cellular metabolism. In diabetes, chronic high blood sugar levels and mitochondrial dysfunction lead to increased production of ROS, further worsening oxidative stress. The excessive accumulation of ROS can damage various cellular components, including proteins, lipids, and DNA, leading to cellular dysfunction and disruption of normal physiological processes [10]. This damage may trigger inflammation and impair the function of essential cellular structures, ultimately contributing to the onset and progression of several diseases, including cardiovascular disease (CVD). Although oxidative stress is a secondary incidence of hyperglycemia in diabetes, it exacerbates the progression of diabetes and its complications [11,12,13]. Oxidative stress has been shown to impair two major mechanisms that fail during diabetes: insulin production by the pancreatic β-cells and tissue-specific insulin signaling [14]. Oxidative stress in diabetes plays a dual role. It not only contributes to the onset of diabetes but also deteriorates the condition and its associated complications. Experimental evidence highlights the involvement of ROS in the impaired function of β-cells, which is caused by autoimmune reactions, cytokines, and inflammatory proteins in type 1 diabetes [15]. Additionally, hyperglycemia has been observed to promote oxidative stress through the generation of free radicals and the suppression of antioxidant defense systems [16]. In chronic hyperglycemia, the production of ROS is sustained, leading to a significant reduction in antioxidant enzymes and non-enzymatic antioxidants across various tissues [17]. This exacerbates oxidative stress, which explains why individuals with diabetes tend to have higher levels of oxidative stress compared to healthy individuals. This review discussed the sources, and the molecular mechanisms involved in the initiation of oxidative stress in diabetes, as well as the role of oxidative stress in major macro and microvascular complications in diabetes, including atherosclerosis, DR and DKD.

## 2. Sources of ROS in DM

### 2.1. Mitochondria

Superoxide anion is mainly produced in the mitochondria, particularly at complexes I and III of the electron transport chain (ETC) [18]. This production occurs when electrons leak from the ETC, resulting in the partial reduction of molecular oxygen into superoxide instead of water [19]. Mitochondrial ROS (mtROS) can also disrupt the function of ETC complexes by oxidizing their iron-sulfur centers, which further elevates ROS production [20,21]. At physiological levels, ROS serve critical roles as signaling molecules that regulate important processes, including cell growth, differentiation, senescence, apoptosis, and autophagy [8,22]. However, when present at pathological levels, ROS can cause cellular dysfunction, cell death, and organ failure [23]. Numerous studies have highlighted the strong link between ROS generation in mitochondria, endothelial dysfunction, and heightened cardiovascular risk [24,25]. The production of ROS promotes the accumulation of leukocytes in vessel walls through processes like rolling, adhesion, and transmigration across the endothelial barrier [26]. In diabetic conditions, blood polymorphonuclear leukocytes (PMNs) demonstrate increased mtROS production [27], reduced oxygen consumption [28], and decreased rolling velocity [29], which correlates with an enhanced interaction with endothelial cells. This interaction is one of the early steps leading to endothelial damage [30]. Excessive mtROS production from the ETC is linked to the hyperglycemic damage seen in diabetes [16,31]. In contrast, inhibiting complex II in bovine aortic endothelial cells (BAECs) has been shown to block glucose-induced activation of protein kinase C (PKC), activation of NFĸB, and formation of AGEs [32]. While the ETC complexes are the primary sources of mtROS, other proteins can also induce mitochondrial oxidative stress, including p66^Shc^, a 66 kDa adaptor protein that belongs to the ShcA protein family [33]. It transmits pro-apoptotic signals by generating mtROS and can contribute to the regulation of lifespan in mammals [34]. A portion of p66^Shc^ is located in the mitochondrial intermembrane space [35], where it acts as a redox enzyme, generating ROS through the oxidation of cytochrome c [36]. This process also results in the partial reduction of molecular oxygen to superoxide. Overexpression of p66^Shc^ leads to endothelial dysfunction, characterized by increased ROS production, elevated levels of E-selectin, and enhanced leukocyte transmigration across the human umbilical vein endothelial cell (HUVEC) monolayer [37]. Additionally, elevated p66^Shc^ levels have been implicated in circulating leukocytes from diabetic patients, as well as in the aorta and renal cortex of diabetes models [38,39]. The p66^Shc^ protein is also implicated in β-cells dysfunction and insulin resistance caused by saturated fatty acids and excessive body fat [40]. Silencing p66^Shc^ gene has been shown to reduce ROS production, enhanced endothelium-dependent vasorelaxation, and diminished apoptosis by restricting cytochrome c release, caspase 3 activity, and poly (ADP-ribose) polymerase cleavage [41]. In contrast, knocking down p66^Shc^ in endothelial cells from diabetic mice reduces ROS production, prevents the formation of the AGE precursor methylglyoxal [42], and protects against ROS-mediated atherosclerosis, diabetes-induced endothelial dysfunction, and glomerulopathy [39]. Therefore, targeting p66^Shc^ may be a promising approach to reduce mtROS production and subsequent EC dysfunction.

### 2.2. NADPH Oxidase

The NADPH oxidases (NOX), consisting of seven membrane-bound enzyme complexes, reduce molecular oxygen to superoxide using NADPH as the electron donor [43,44]. Their primary role is to generate ROS, contributing to the ROS burst and bacterial elimination in phagocytes [32,45,46]. Among the isoforms, NOX4 is most prevalent in endothelial cells compared to NOX1, NOX2, and NOX5 [47]. In non-phagocytic cells under normal physiological conditions, moderate levels of ROS are produced by the catalysis of NOX enzymes, essential for typical redox signaling [48]. Specifically, NOX2 and NOX4 promote endothelial cell proliferation and survival by ROS-mediated activation of p38 MAPK and Akt pathways [49,50]. The significant decrease in ROS-mediated tube formation and wound healing in human microvascular endothelial cells (HMECs) after siRNA-mediated NOX4 knockdown further underscores its crucial role in angiogenesis and endothelial cell proliferation [51]. Conversely, in pathological states, NOX4 might lead to a pro-thrombogenic endothelial phenotype. A study demonstrated that siRNA-mediated NOX4 knockdown reduced the endotoxin-induced expression of intercellular adhesion molecule-1 (ICAM-1), interleukin-8 (IL-8), and monocyte chemoattractant protein-1 (MCP-1) in human aortic endothelial cells (HAECs) [52]. Moreover, elevated NOX4 levels and subsequent augmentation of oxidative stress are associated with idiopathic pulmonary fibrosis [53,54] and the progression of diabetic nephropathy in both humans and animal models [55]. Nonetheless, NOX4 is implicated to have protective effects on the vasculature by preventing endothelial dysfunction in ischemic or inflammatory damage [56]. Furthermore, overexpressed endothelial NOX4 has been shown to improve vasodilation and lower blood pressure in transgenic mice through the generation of H_2_O_2_ [46,57], whereas NOX4 knockout mice displayed worsened angiotensin II-induced inflammation, endothelial dysfunction, and vascular remodeling [56]. Like NOX2 and NOX4 isoforms, NOX1 is also expressed in endothelial cells, but at a lower level [58]. Endothelial NOX1 is implicated in ROS generation under pathological conditions as evident by increased NOX1 expression in endothelial cells subjected to oscillatory shear stress [59]. Furthermore, siRNA-mediated NOX activator protein 1 knockdown has been shown to decrease ROS generation in HUVECs exposed to oxidized low-density lipoproteins (Ox-LDL) [60]. Unlike the other NOX isoforms, the activity of NOX5 is regulated by intracellular calcium ion (Ca^2+^) levels due to its four Ca^2+^ binding sites in the N-terminal calmodulin-like domain with elevated NOX5 expression is implicated in CAD patient arteries [61]. Studies in animals showed that the NOX1, NOX2, and NOX5 isoforms augment endothelial dysfunction, inflammation, and apoptosis in animal models of induced hypertension [62,63], diabetes [64], or atherosclerosis [62].

### 2.3. Uncoupled Endothelial Nitric Oxide Synthase

Nitric oxide synthases (NOS) are a group of enzymes responsible for producing NO and citrulline from oxygen and L-arginine. This process involves the transfer of electrons from NADPH, which is attached to the C-terminal reductase domain, to the heme iron and the cofactor tetrahydrobiopterin (BH_4_) found in the N-terminal oxygenase domain. This transfer is essential for the reduction and incorporation of O_2_ into L-arginine, resulting in the formation of NO and citrulline [32,65]. NO is generated by three distinct isoforms of NOS, including neuronal NOS (nNOS), inducible NOS (iNOS), and endothelial NOS (eNOS), the latter being the most prevalent in the endothelium [66]. NO generated by eNOS plays a crucial role in maintaining vascular homeostasis [67]. However, in unhealthy physiological conditions, including diabetes or other metabolic abnormalities, a scarcity of substrates or cofactors for eNOS may produce superoxide instead of NO, a phenomenon referred to as “uncoupling” [68]. Additionally, NO can react with superoxide to form peroxynitrite, a strong oxidant that leads to protein nitration, mitochondrial dysfunction, and damage or death of endothelial cells [69,70,71,72]. A decrease in BH_4_ bioavailability appears to be the primary factor driving eNOS uncoupling linked to vascular injury [32,46,66,67]. In ROS-induced endothelial dysfunction, BH_4_ is converted to BH_2_, which cannot serve as a cofactor for eNOS, contributing to the uncoupling [73]. Studies have shown that knocking out p47phox, a subunit of NOX, as well as the inhibition of another subunit, Rac1, have been shown to prevent both eNOS uncoupling and BH_4_ oxidation, highlighting the critical function of NOX in decreasing BH_4_ availability and facilitating eNOS uncoupling. Additionally, studies in humans demonstrated that endothelial function is significantly improved in patients with diabetes [74], hypertension [75], or hypercholesterolemia [76] after BH_4_ supplementation. An endogenous competitive inhibitor, asymmetric dimethylarginine (ADMA), is implicated in NOS uncoupling, as evident in pulmonary arterial endothelial cells, where it promotes eNOS uncoupling [77]. It has been reported that in patients with advanced atherosclerosis, eNOS uncoupling and O_2_· production within the vascular endothelium is positively correlated with Serum ADMA levels [78]. Multiple in vitro studies showed that when endothelial cells are exposed to various stimuli mimicking metabolic changes leading to endothelial dysfunction, such as native LDL and Ox-LDL [79], angiostatin [80], homocysteine [81], and high glucose [82,83], they become increasingly vulnerable to eNOS uncoupling. eNOS uncoupling is implicated in promoting the mobilization and function of endothelial progenitor cells (EPCs). For instance, ROS levels significantly augmented in EPCs obtained from diabetic patients compared to control subjects, whereas the inhibition of eNOS attenuated this ROS elevation. Additionally, high glucose treatment significantly decreased BH_4_ levels and altered migration capability in the cultured EPCs, which were restored by eNOS inhibition and the administration of exogenous BH_4_ [84].

### 2.4. Xanthine Oxidase

Xanthine oxidase (XO) belongs to the xanthine oxidoreductase (XOR) family, which play a vital role in purine degradation by catalyzing the conversion of hypoxanthine to xanthine and later xanthine to uric acid. This enzyme operates as a homodimer, around 300 kDa in size, with each monomer containing a molybdopterin (Mo-Pt) cofactor, two iron-sulfur (Fe-S) centers, and a flavin adenine dinucleotide (FAD) domain [85,86]. Typically, expressed in its dehydrogenase form (XDH), XOR can switch to its oxidase form (XO) under inflammatory conditions due to the oxidation of cysteine residues 535 and 992 or through proteolytic conversion [85,87]. In the process of converting xanthine into uric acid, XDH facilitates the reduction of NAD+ to NADH, whereas the XO form reduces molecular oxygen to superoxide (O_2_·) and H_2_O_2_. XO-induced generation of ROS is the primary contributor to oxidative stress in ischemia/reperfusion injuries. Pharmacological Inhibition of XO with allopurinol is shown to decrease vascular remodeling and oxidative stress in a rodent model of hypoxia-induced pulmonary hypertension [88]. Additionally, XO is associated with elevated ROS levels and vascular damage in diabetes, as elevated XO levels are detected in the plasma of type 1 [89] and type 2 [90] diabetic patients. Furthermore, allopurinol administration reduced plasma lipoperoxides in those with type 1 diabetes and attenuated increased superoxide levels in the aortic rings of diabetic rabbits [89]. Moreover, a connection between elevated XO levels and atherosclerosis was established. Bovine aortic endothelial cells (BAECs) subjected to oscillatory shear stress exhibited increased superoxide production and XO activity, which was suppressed by the XO inhibitor oxypurinol [91]. Notably, increased oxidative stress-induced activation of NOX leads to XO production [92]. Apocynin-mediated NOX inhibition blocked superoxide generation and the conversion of XDH to XO in BAECs under shear stress. Additionally, BAECs treated with AngII showed higher XO levels, an effect negated by NOX inhibition. Importantly, under certain pathological circumstances, including liver and intestinal diseases [93] and hypoxia [94] XDH is released into the bloodstream and transformed into the XO form, which binds to sulphated glycosaminoglycans (GAGs) expressed on endothelial cell surfaces, thereby further promoting ROS generation and endothelial dysfunction [95]. It is noteworthy that inhibiting circulating XO activity enhanced endothelial function in patients suffering from CAD [92], chronic heart failure [96], as well as type 1 DM [89] and T2DM [97] diabetes.

### 2.5. Nutrient Excess

In cells, overproduction of ROS often occurs when there is an excessive nutrient supply, such as high glucose or elevated free fatty acids, combined with low energy demand. This state leads to elevated intracellular ATP levels, which inhibit ADP availability and slow electron flow along the ETC in mitochondria. When the ETC is slowed, the electron carriers, particularly at Complexes I and III, remain in a reduced state for longer, increasing the likelihood of electrons leaking directly to molecular oxygen, thereby forming superoxide anion (O_2_^−^•) as a primary ROS. In hyperglycemia, increased glycolytic flux leads to an abundance of pyruvate entering the tricarboxylic acid (TCA) cycle, generating excess NADH and FADH_2_, which further supply electrons to the ETC and exacerbate the reduced state of its components. Similarly, elevated free fatty acids in obesity undergo β-oxidation, producing additional NADH and FADH_2_, contributing to the electron overload in mitochondria. The resulting mitochondrial ROS production can damage mitochondrial DNA, proteins, and lipids, further impairing ETC function and creating a vicious cycle of ROS generation. In endothelial cells, this oxidative environment reduces NO bioavailability by reacting with NO to form peroxynitrite, leading to endothelial dysfunction, vasoconstriction, and inflammation. These processes collectively link excess nutrient-induced metabolic conditions, such as hyperglycemia and obesity, to mitochondrial ROS overproduction and subsequent vascular dysfunction, which are key contributors to the development of diabetic vascular complications [98]. Oxidative stress often arises early in response to rapid or sustained hyperglycemia. Several cellular changes occur in response to oxidative stress from hyperglycemia, including: (i) generation of peroxynitrites; (ii) increased levels of ADMA, a competitive eNOS inhibitor; (iii) reduced insulin-induced generation of NO [99]; (iv) Augmented diacylglycerol (DAG) synthesis followed by PKC activation, which are linked to NOX activation, reduced insulin-stimulated NO production [100], upregulated expression of adhesion molecules, increased endothelin-1 (ET-1) release [101], and p66Shc activation [102,103]; (v) decreased activity and/or expression of antioxidant enzymes [104]; (vi) formation of AGEs [105]; and (vii) activation of protein phosphatase 2A (PP2A) [106]. AGEs possess pro-inflammatory characteristics created when reducing sugars react with amino groups in proteins, lipids, and nucleic acids through a non-enzymatic Maillard reaction. AGEs can be generated through other mechanisms, including glucose oxidation, lipid peroxidation, or the polyol pathway [107,108,109], accumulating in blood vessels and contributing to the pathophysiology of both microvascular and macrovascular complications associated with diabetes. For instance, patients with Type 2 diabetes exhibit elevated levels of compounds like glyoxal, methylglyoxal, and 3-deoxyglucosone [110]. AGEs can exacerbate diabetic complications by impairing cellular structures, binding specific basement membrane molecules in the extracellular matrix, and interacting with receptors for AGEs (RAGEs) on cell surfaces, relaying stress signals [111]. In endothelial cells, the interaction between AGEs and RAGEs boosts ROS production and increases adhesion molecules, including vascular cell adhesion molecule-1 (VCAM-1), ICAM-1, and E-selectin [111,112], resulting in NO bioavailability due to reduced eNOS activity, while promoting cell apoptosis [113,114,115]. A study demonstrated that cultured endothelial cells exposed to AGE methylglyoxal (MGO) significantly reduced the Bcl2/Bax ratio, leading to increased apoptosis. This was associated with an augmented production of ROS by NOX4, diminished eNOS activity, mitochondrial membrane potential, and nuclear translocation of NFκB, all of which could be reduced by pretreating endothelial cells with phosphocreatine [115]. Another potential mechanism of high glucose-induced ROS production is the increased expression and activation of PP2A, a serine/threonine phosphatase that dephosphorylates various substrates involved in cellular signaling, including p66^Shc^. PP2A catalyzes the dephosphorylation of p66Shc at Ser^36^ and promotes its translocation into mitochondria to induce oxidative stress [103]. A study showed that high glucose-treated HUVECs significantly augmented intracellular calcium levels, enhanced the phosphorylation of Ca^2+^/calmodulin-dependent protein kinase II (CaMKII) and cAMP response element-binding protein (CREB), ultimately upregulated both PP2A expression and activity, as well as ICAM-1 expression. However, blocking the high glucose-induced expression of PP2A prevented O_2_· accumulation, enhanced NO production, and inhibited ICAM-1 expression, thereby exacerbating endothelial dysfunction [116]. Likewise, increased lipid levels contribute to endothelial cell dysfunction and death due to increased oxidative stress. For instance, increased levels of palmitate, the primary saturated fatty acid in human plasma [117,118], significantly disrupt insulin signaling in endothelial cells. Specifically, treating bovine aortic endothelial cells (BAECs) with palmitate significantly diminished insulin-mediated tyrosine phosphorylation of insulin receptor substrate-1 (IRS-1) and the subsequent serine phosphorylation of Akt and eNOS, along with decreased NO production, due to increased inhibitor of NFκB kinase beta (IKKβ) activity and subsequent phosphorylation of IRS-1 that inhibited the signaling pathway [118]. Another study reported that palmitate-treated HMECs augmented NOX4-induced ROS production, NFκB activation, and upregulation of IL-6 and ICAM-1 expression [119]. Furthermore, chronic exposure to palmitate resulted in apoptosis of endothelial and progenitor cells through the activation of stress-activated protein kinases JNK and p38 MAPK [120,121,122,123]. Collectively, these findings suggest that increased plasma levels of saturated fatty acids can induce endothelial dysfunction and apoptosis while impairing endothelial repair by damaging endothelial progenitor cells. Importantly, elevated levels of saturated fatty acids promote the formation and accumulation of detrimental lipid metabolites like ceramide [124], which is known to induce superoxide production and peroxynitrite formation through NOX in bovine coronary arterial endothelial cells [125], while declining bioactive NO levels in human endothelial cells [126]. In contrast, inhibiting ceramide synthesis in fat-fed streptozotocin (STZ)-treated rats has been shown to improve endothelial dysfunction by enhancing phosphorylation and NO release via the PI3K/Akt/eNOS pathway.

### 2.6. Peroxiredoxins

Peroxiredoxins (Prx) are thiol-specific enzymes that convert H_2_O_2_ into water by utilizing active cysteine residues. In mammals, six isoforms of peroxiredoxins (Prx1–6) have been characterized so far. Isoforms Prx1–5 require two cysteine residues for function, whereas Prx6 requires only one [127]. Multiple studies revealed the essential role of Prx as ROS scavengers, particularly in protecting lung tissues from ROS-induced cytotoxicity [128,129,130]. Silencing Prx2 with siRNA resulted in the inactivation of the vascular endothelial growth factor receptor-2 (VEGFR2) due to H_2_O_2_-induced oxidation of a crucial cysteine residue and subsequently reduced chemotactic mobility, proliferation, and VEGF-induced tube formation of HAECs. Interestingly, VEGF-induced VEGFR2 activity was not altered upon knockdown of Prx1 and Gpx1, indicating that Prx2 is the specific cytosolic antioxidant enzyme responsible for regulating basal H_2_O_2_ levels in endothelial cells [131]. Prx6 KO mice exhibited dysfunctional and apoptotic endothelial cells around skin wounds, which were correlated with damage to blood vessels and hemorrhages. Moreover, the siRNA-mediated knockdown of Prx6 significantly decreased the survival of endothelial cells subjected to H_2_O_2_ exposure [132]. Prx1 isoform plays an important role in the antioxidative and anti-inflammatory effects in BAECs upon laminar shear stress [133].

### 2.7. Thioredoxin

Thioredoxins (Trxs) are a group of 12 kDa oxidoreductases found in the cytosol and mitochondria. They are essential for reducing disulfides and sulfenic acids that develop from oxidative stress in proteins, both within and between cells. For instance, Trx possesses cysteine residues in Prx in a reduced state, preserving the enzyme’s catalytic activity [32]. Beyond the regulation of redox conditions of various targets, Trx aids the survival of endothelial cells. In these cells, H_2_O_2_ has been reported to regulate Trx protein levels. For instance, H_2_O_2_ at low concentrations elevates Trx levels and shields against apoptosis of endothelial cells, whereas higher concentrations augment apoptosis through Trx degradation via a cathepsin-D dependent mechanism [134]. Trx also affects endothelial function by binding to specific proteins and transcription factors. Laminar shear stress is implicated to augment the activity of Trx due to the decreased expression of thioredoxin binding protein (TXNIP), facilitating Trx’s binding with apoptosis signal-related kinase-1 (ASK1) following the inhibition of ASK1-mediated activation of JNK and p38 MAPK in response to TNF [135]. Additionally, in HUVECs, the prevention of apoptosis from A low dose of H_2_O_2_ induces the translocation of Trx into the nucleus, where it binds to several transcription factors, improving their interaction with antioxidant responsive elements (AREs) and promoting the expression of glutathione S-transferase P1, which is crucial for preventing apoptosis [136]. Beyond Trx, other proteins can also contribute to the buffering of ROS. For example, Paraoxonase 2 (PON2) binds to coenzyme Q10 within respiratory complex III, decreasing ROS production at that site [137]. PON2 KO mice have shown diminished activity in complexes I and III, along with lower oxygen consumption and ATP production. Additionally, these mice display a more severe atherogenic phenotype than controls when fed a high-fat, high-cholesterol diet [138].

### 2.8. Uncoupling Proteins

Uncoupling proteins (UCPs), which consist of five mitochondrial protein carriers, transport protons from the intermembrane space to the matrix, reducing the mitochondrial membrane potential followed by uncoupling ATP synthesis. Given the necessity of high levels of mitochondrial membrane potential for ROS production, uncoupling proteins may serve as a form of antioxidant defense. Indeed, overexpressing UCP1 has been shown to diminish ROS production and apoptosis in endothelial cells triggered by high glucose levels [32,139], while UCP2 knockdown augmented mitochondrial membrane potential and superoxide anion levels in murine endothelial cells [140]. Hyperglycemia-induced upregulation of UCP2 is implicated in endothelial cells [141]. Animal studies showed that UCP2 upregulation enhances endothelium-dependent relaxation in aortic rings from diabetic mice [142], as well as prevents endothelial cell apoptosis resulting from elevated free fatty acids and ROS [143].

### 2.9. Superoxide Dismutases

Superoxide dismutases (SODs) are a group of enzymes that facilitate the transformation of superoxide anion into H_2_O_2_. There are three distinct isoforms of SOD located in various subcellular compartments: a cytosolic copper-zinc SOD (CuZnSOD), a manganese SOD (MnSOD) predominantly found in mitochondria, and an extracellular CuZnSOD (SOD3) that binds to cell surface heparin sulfate proteoglycans [144,145]. Inhibition of SOD1 in endothelial cells augmented superoxide production and diminished the phosphorylation of extracellular signal–regulated kinases (ERK) 1/2 mediated by fibroblast growth factor-2 (FGF-2) and vascular endothelial growth factor (VEGF), thereby decreasing angiogenesis [146]. Beyond its traditional role in scavenging ROS, H_2_O_2_ produced by SOD1 has been identified as an endothelium-dependent hyperpolarization factor in vivo [147]. SOD2, an inducible enzyme featuring manganese at its active site, catalyzes superoxide generation in the mitochondrial matrix [148]. SOD2-mediated generation of H_2_O_2_ has been shown to promote endothelial cell sprouting and angiogenesis [149]. Moreover, SOD2 expression levels are significantly higher in human endothelial progenitor cells relative to differentiated endothelial cells, making them more resilient to oxidative stress [150]. Elevated superoxide levels and impaired relaxation of aortas in response to acetylcholine have been observed in experimental atherosclerosis models with SOD2 deficiency [151]. Furthermore, decreased SOD2 levels were detected in the pulmonary artery endothelial cells of fetal lambs with chronic pulmonary hypertension [152]. DM-induced ROS production also impairs SOD2 expression and activity, thereby exacerbating ROS-mediated endothelial dysfunction. Notably, a positive correlation was found between decreased SOD2 levels and impaired wound healing and angiogenesis in endothelial progenitor cells isolated from diabetic mice [153]. SOD3 serves as the predominant SOD in the vascular extracellular space [154]. Studies demonstrated that decreased SOD3 levels are linked to elevated ROS levels in the aortas of older rats [155], while upregulated expression of SOD3 enhanced endothelial function in hypertensive [156] and heart failure rat models [157].

### 2.10. Catalase and Peroxidases

Antioxidant enzymes, catalase and peroxidases, catalyze the conversion of H_2_O_2_ into molecular oxygen and water. Catalase is a 240-kDa homotetrameric heme-containing peroxisomal protein, predominantly expressed in the liver, lungs, and kidneys [158]. Although catalase’s role in the endothelium is not well-known as its activity is not well studied in normal physiological conditions, in endothelial cells, it plays an active role in the adaptive response of cells against oxidative stress and may be modulated by oxidative factors like oxLDL [159]. Glutathione peroxidase (GPx) is an 85-kDa selenium-dependent protein, exerts its catalytic activities using monomeric glutathione as an electron donor. The oxidized form of glutathione is then reverted to its reduced state by the action of glutathione reductase. Four types of glutathione peroxidases have been identified so far in mammals, with isoform 1 (GPx-1) being the most abundantly expressed [160]. In the endothelial cells, GPx-1 is present in both the mitochondria and cytoplasm [161]. Upregulation of GPx-1 expression is implicated in protecting cultured human primary pulmonary artery endothelial cells from H_2_O_2_-induced cytotoxicity [162]. In contrast, the knockdown of GPx-1 enhances leukocyte adhesion to endothelial cells and promotes a pro-inflammatory phenotype in aging [163]. Furthermore, the mesenteric artery of GPx-1 KO mice demonstrated impaired vasodilation due to reduced NO bioavailability and elevated oxidative stress [160]. MicroRNAs play an important role in regulating genes involved in the antioxidant response to elevated glucose levels; for instance, a study revealed that upregulation of miRNA-185 coincided with the suppression of GPx-1 levels in HUVECs exposed to fluctuating high glucose [164]. Possible sources of ROS are illustrated in Figure 1.

## 3. Mechanism of Oxidative Stress in DM

In diabetes, oxidative stress results from a complex interplay of multiple factors, such as the buildup of glycolysis intermediates, formation of advanced glycation end products (AGEs), activation of PKC, polyol, and hexosamine pathways [17] (Table 1).

### 3.1. Buildup of Glycolysis Intermediates

Under typical physiological conditions, cellular processes, including glucose oxidation, lead to the mitochondrial production of superoxide anion radicals. This production occurs at a level manageable by the body’s antioxidant defenses [18,176]. However, in hyperglycemic conditions, the overproduction of superoxide anion radicals can overwhelm these antioxidant systems, resulting in oxidative stress that damages nuclear DNA and other biomolecules [177]. In response to DNA damage, the DNA repair enzyme poly-ADP-ribose polymerase-1 (PARP-1) becomes activated [178]. This enzyme inhibits GAPDH, causing levels of Glyceraldehyde 3-phosphate (G3P) and other glycolytic intermediates such as fructose-6-phosphate and glucose-6-phosphate, as well as glucose, to rise [179]. The accumulation of these molecules triggers additional pro-oxidative pathways like the AGE and PKC pathways from increased G3P, and the hexosamine and polyol pathways due to elevated F-6-P and glucose levels, respectively [17]. Additionally, the buildup of G3P can lead to its autooxidation, which produces hydrogen peroxide (H_2_O_2_) and exacerbates oxidative stress. Likewise, the accumulation of glucose can result in its autooxidation, yielding glyoxal, a precursor to AGEs, which further contributes to cellular oxidative stress [10].

### 3.2. AGEs/RAGE Pathway

AGEs are highly reactive and irreversible end products resulting from non-enzymatic reactions involving glucuronyl groups and free amino groups, including those found in lipids and proteins [180]. Prolonged hyperglycemia significantly accelerates the production of AGEs, while excessive AGEs directly enhance the production of ROS. Importantly, the generated ROS reciprocally stimulates further AGE production, thereby exacerbating oxidative stress damages to different cells in different organs. A substantial body of research indicates that the receptor for AGEs (RAGE) is increasingly expressed in vascular endothelial cells, immune cells, monocytes/macrophages, neurons, cardiomyocytes, adipocytes, glomerular epithelial cells, podocytes and alveolar epithelial cells following hyperglycemic stimulation [167]. In endothelial cells, the interaction of RAGE with AGEs activates NOX, resulting in increased ROS production, which disturbs molecular conformation and alters enzyme activity, thereby inducing oxidative stress responses. Consequently, the activated oxidative stress augments the activation of different downstream signaling pathways, including NFκB, TNF, JNK, and p38-MAPK, leading to an increased release of adhesion molecules, vascular endothelial growth factors, and inflammatory factors [181,182].

### 3.3. PKC Pathway

PKC pathway plays a pivotal role in the cellular response to oxidative stress. The generation of ROS induced by oxidative stress directly activates various PKC isoforms through oxidative modifications of their regulatory domains [183]. In diabetes, hyperglycemia causes the intracellular diacylglycerol content to significantly increase, leading to the activation of the PKC pathway [184,185]. Furthermore, AGE/RAGE and polyol pathways can activate the PKC.

Upon activation, PKC may either promote cellular survival or instigate apoptosis, contingent upon the specific isoform and the cellular context. For instance, PKC-ε has been demonstrated to enhance antioxidant defenses by activating protective pathways, such as the Nrf2-mediated antioxidant response [186], whereas [187]. Furthermore, PKC can exacerbate oxidative stress by augmenting ROS production through the stimulation of NOX activity [188]. Through these diverse mechanisms, the PKC pathway serves as a critical mediator in determining cellular outcomes during oxidative stress, thereby influencing the development and progression of diseases, including cardiovascular disorders, neurodegenerative diseases, complications related to diabetes, and cancer [189].

### 3.4. Polyol Pathway

The polyol pathway plays a significant role in oxidative stress, particularly during chronic hyperglycemia, as seen in DM. Within this pathway, excess intracellular glucose is converted into sorbitol by the enzyme aldose reductase, which uses NADPH as a cofactor [190]. Sorbitol is then oxidized to fructose by sorbitol dehydrogenase, a reaction that produces NADH [166]. The activation of the polyol pathway under hyperglycemic conditions leads to several detrimental effects. Firstly, the consumption of NADPH reduces its availability for other vital antioxidant systems, especially for regenerating reduced glutathione (GSH), crucial for neutralizing ROS [191]. Secondly, the elevated NADH/NAD^+^ ratio from increased sorbitol dehydrogenase activity enhances mitochondrial superoxide production, worsening oxidative stress [192]. Additionally, sorbitol, which does not easily diffuse across cell membranes, accumulates within cells, leading to osmotic stress and subsequent damage to cellular structures [193]. Collectively, these changes create a cycle of oxidative damage and inflammation, significantly contributing to the development of diabetic complications like neuropathy, retinopathy, nephropathy, and CVD [104]. As a result, aldose reductase inhibitors have been investigated as potential therapies to reduce oxidative damage by regulating polyol pathway activity [194,195]. While experimental findings are promising, challenges remain in clinical applications, highlighting the complexity of oxidative stress mechanisms and the necessity for thorough strategies to address oxidative injury in diabetes [104].

### 3.5. Hexosamine Pathway

The hexosamine biosynthetic pathway (HBP) acts as a nutrient sensor and plays a key role in oxidative stress and cellular dysfunction in diabetes, obesity, and cardiovascular diseases. Normally, only a small fraction (2–5%) of glucose enters the HBP to produce UDP-N-acetylglucosamine (UDP-GlcNAc), which is used for O-linked β-N-acetylglucosamine (O-GlcNAc) modification of proteins in the nucleus, cytoplasm, and mitochondria [196]. Under hyperglycemic conditions, increased glucose flux through the HBP leads to excessive O-GlcNAcylation of regulatory proteins, altering gene expression, impairing mitochondrial function, and increasing oxidative stress [197,198]. For example, over-O-GlcNAcylation of the antioxidant transcription factor Nrf2 reduces its nuclear translocation and target gene activation, weakening antioxidant defenses and promoting ROS accumulation [199]. Additionally, HBP hyperactivation increases endoplasmic reticulum (ER) stress by promoting protein misfolding, which triggers the unfolded protein response and further ROS production [200]. Mitochondrial proteins are also subject to O-GlcNAcylation under high-glucose conditions, resulting in impaired ETC activity, altered mitochondrial dynamics, and elevated mtROS generation [201]. Notably, HBP-induced modification impairs insulin signaling pathways, which contribute to insulin resistance, a condition that serves as both a cause and a consequence of oxidative stress in metabolic tissues. Overall, HBP links metabolic overload to oxidative stress, inflammation, and organ dysfunction. Targeting HBP flux or enzymes regulating O-GlcNAc cycling, such as O-GlcNAc transferase (OGT) and O-GlcNAcase (OGA), holds promise for mitigating oxidative stress-related metabolic diseases.

Having discussed the mechanisms underlying oxidative stress in diabetes, it is important to consider how these processes translate into vascular tissue damage and disease-specific pathophysiology. Persistent oxidative stress in diabetes directly impairs endothelial function, disrupts vascular homeostasis, and promotes inflammatory and fibrotic responses, leading to structural and functional alterations within the vasculature. These oxidative stress-induced changes contribute to the development and progression of both microvascular and macrovascular complications in diabetes, including atherosclerosis, DR, and DKD. In the following sections, we will discuss how oxidative stress drives vascular dysfunction and specific pathophysiological mechanisms underlying these diabetes-related vascular complications.

## 4. Role of Oxidative Stress in Vascular Cells

Regulating vascular tone is essential for maintaining blood vessel homeostasis and ensuring adequate blood flow to peripheral organs [202]. Under physiological conditions, proper endothelial function allows for vasorelaxation through the release of vasoactive substances [203]. However, an imbalance in the production of protective and relaxing factors versus constricting substances by the endothelium indicates endothelial dysfunction, which often precedes various vascular diseases [204]. Oxidative stress plays a pivotal role in both the onset and development of endothelial dysfunction and vascular diseases that affect multiple cell types within the vascular wall [205]. Furthermore, oxidative stress within the vascular system triggers systemic inflammation through immune activation [206]. Immune cells that are activated migrate into blood vessels and release various factors such as ROS, metalloproteinases, cytokines, and chemokines, which contribute to endothelial and vascular smooth muscle cells (VSMCs) dysfunction and subsequent vascular injury, leading to vasoconstriction and remodeling of the vessels [205].

### 4.1. Oxidative Stress and Endothelial Cells

Endothelium is a monolayer of endothelial cells that lines the interior of blood vessels and plays a crucial role in maintaining vascular homeostasis [207,208]. In addition to this, healthy endothelial cells perform several other functions, such as the regulation of vascular tone, maintaining blood fluidity, modulating inflammation and immune responses, and facilitating the formation of new blood vessels (neovascularization) [30,209]. Impairment of the endothelium is a complex pathophysiological phenomenon that involves elevated activation of endothelial cells as well as the onset of endothelial dysfunction [30]. Endothelial activation refers to a pro-inflammatory and pro-coagulant state of these cells, associated with the expression of cell-surface adhesion molecules that are essential for recruiting and attaching inflammatory cells [210,211]. This activation is triggered by cytokines released by tissues and organs with inflammation. Studies reported that oxidative stress plays a significant role in mediating the production and secretion of cytokines, thus linking ROS to inflammation, as well as endothelial activation and dysfunction [212]. Endothelium-derived NO contributes to the maintenance of vascular homeostasis. A decrease in NO bioavailability, which can occur due to reduced NO production or increased degradation by superoxide anions, signals the initiation of endothelial dysfunction. Superoxide anions interact with nitric oxide (NO) to produce peroxynitrite (ONOO^−^) [213,214,215]. Peroxynitrite is implicated in promoting protein nitration, leading to the dysfunction and apoptosis of endothelial cells [70,71,72]. NOX, xanthine oxidase, and uncoupled eNOS contribute to the production of superoxide anions. Furthermore, impairment in mitochondrial respiratory chain complexes significantly increases the production of oxidative stress-related substances. Pathobiological and metabolic factors, such as hyperglycemia, hyperlipidemia, hypertension [216], mental stress [217], aging [218], and drug exposure [219] can impair endothelial function by disturbing the molecular mechanisms behind NO generation. As a result, therapeutic interventions that target improving insulin sensitivity, glycemic control, lipid levels, and blood pressure can frequently mitigate endothelial dysfunction. Additionally, epigenetic factors such as histone acetylation and deacetylation, as well as modulation of microRNA expression, are implicated in influencing vascular homeostasis. The role of oxidative stress in the vascular endothelial cell is illustrated in Figure 2.

### 4.2. Oxidative Stress and Smooth Muscle Cells

Oxidative stress plays a critical role in the pathophysiology of vascular smooth muscle cells (VSMCs), which are key players in the maintenance of vascular tone, structure, and remodeling. In normal physiological conditions, VSMCs maintain a contractile phenotype essential for blood pressure regulation and vascular homeostasis [220]. However, under oxidative stress, there is a well-documented phenotypic switch from a contractile to a synthetic phenotype. This transformation is associated with increased VSMC proliferation, migration, extracellular matrix production, and a pro-inflammatory profile, all of which are implicated in vascular pathologies such as atherosclerosis, hypertension, and restenosis following vascular injury [221]. The major sources of ROS in VSMCs, include NOX, mitochondrial respiration, xanthine oxidase, and uncoupled eNOS [222]. Among these, the NOX family, particularly NOX1 and NOX4, are highly expressed in VSMCs and have been shown to mediate numerous redox-sensitive signaling pathways involved in vascular remodeling [223]. Elevated ROS levels influence multiple cellular processes by modifying redox-sensitive proteins and signaling molecules. For instance, ROS-induced activation of mitogen-activated protein kinases (MAPKs) and NFκB facilitates the expression of pro-inflammatory cytokines and matrix metalloproteinases (MMPs), which degrade the extracellular matrix and contribute to plaque instability in atherosclerosis [224]. Furthermore, oxidative stress is linked to the activation of transcription factors such as activator protein-1 (AP-1) and hypoxia-inducible factor-1α (HIF-1α). A recent study demonstrated that high-salt conditions enhance VSMC proliferation and migration by upregulating HIF-1α, which in turn increases oxidative stress and promotes vascular remodeling [225]. This indicates a feedback loop wherein oxidative stress both induces and is amplified by phenotypic switching of VSMCs. Another crucial consequence of oxidative stress is the induction of apoptosis in VSMCs, particularly in the context of aortic dissection and aneurysm formation. Apoptotic loss of VSMCs compromises the structural integrity of the vascular wall, rendering it prone to rupture. A study demonstrated that oxidative stress contributes to VSMC apoptosis via the mitochondrial pathway, with increased Bax/Bcl-2 ratios and cytochrome c release into the cytosol. This apoptotic cascade, if unregulated, leads to thinning of the medial layer and predisposes vessels to catastrophic events such as dissection [226]. Moreover, oxidative stress-induced apoptosis often coexists with the dysregulation of autophagy, which further compromises cellular survival and vascular homeostasis [227]. VSMCs are also affected by metabolic alterations driven by oxidative stress. Hyperuricemia, for instance, has emerged as a metabolic trigger that increases ROS production in VSMCs. A recent study reported that uric acid elevates intracellular oxidative stress by disrupting mitochondrial function, leading to decreased nitric oxide bioavailability and suppression of tumor suppressor p53, which is vital for maintaining VSMC quiescence and genomic stability [228]. These findings are particularly relevant in metabolic syndrome and gout, where elevated uric acid levels are common and contribute to vascular dysfunction through redox imbalance. The crosstalk between oxidative stress and other forms of cellular stress, such as endoplasmic reticulum (ER) stress, is also an area of increasing interest. ER stress occurs when misfolded proteins accumulate within the ER lumen, triggering the unfolded protein response (UPR). In oxidative stress, ROS can exacerbate ER stress, leading to VSMC inflammation and death. A recent study showed that hypertensive patients exhibit increased markers of both oxidative and ER stress in their vascular tissue, suggesting a synergistic pathogenic mechanism [229]. The convergence of these stress responses amplifies cellular damage and accelerates vascular disease progression. From a therapeutic perspective, targeting oxidative stress in VSMCs offers promising avenues for the prevention and treatment of vascular diseases. Antioxidant strategies include both pharmacological agents and natural compounds. Ginsenoside Rb1, a major component of ginseng, has been shown to exert vascular protective effects by modulating oxidative stress and inflammation in VSMCs. A study by Lu et al. demonstrated that Rb1 inhibits NOX4 expression and reduces ROS generation, thereby preventing VSMC phenotypic transformation and migration [230]. Another promising strategy involves activation of the Mas receptor (MasR) and particulate guanylyl cyclase A (pGCA), which enhance antioxidant responses in VSMCs. In a preclinical study, MasR and pGCA activation improved VSMC survival and attenuated ROS-mediated injury, suggesting their potential as drug targets [231]. Moreover, gene therapy approaches aiming to modulate redox-sensitive signaling molecules are under investigation. For instance, the delivery of catalase or SOD genes to vascular tissues has shown efficacy in reducing oxidative stress and mitigating VSMC dysfunction in animal models. Additionally, pharmacological inhibitors of NOX have been developed, with NOX1-specific inhibitors like GKT137831 demonstrating favorable effects in reducing oxidative damage and vascular inflammation [232]. However, clinical translation of these agents remains a challenge due to issues related to bioavailability, specificity, and off-target effects. In conclusion, oxidative stress plays a multifaceted and central role in the regulation of vascular smooth muscle cell function in normal physiology and pathological states. Through mechanisms involving phenotypic modulation, inflammation, apoptosis, and interaction with other cellular stress responses, ROS contribute to the pathogenesis of various vascular disorders. A deeper understanding of the redox signaling pathways in VSMCs is essential for the development of targeted therapeutic interventions. As research progresses, integrating redox biology into the clinical management of cardiovascular diseases may yield novel strategies to combat vascular dysfunction and improve patient outcomes.

### 4.3. Oxidative Stress and Vascular Fibroblast

Oxidative stress plays a critical role in the pathophysiology of vascular fibroblasts, influencing their behavior, phenotype, and contribution to cardiovascular diseases. Vascular fibroblasts, traditionally seen as passive structural components of the vessel wall, are now recognized as dynamic cells capable of responding to environmental stimuli, including ROS [233]. In vascular fibroblasts, oxidative stress activates redox-sensitive transcription factors such as NFκB, AP-1, and Nrf2, which in turn upregulate pro-inflammatory cytokines, MMPs, and adhesion molecules [46]. These molecules mediate fibroblast activation and transition into a myofibroblast phenotype, characterized by increased contractility, secretion of extracellular matrix (ECM) components such as collagen types I and III, and expression of α-smooth muscle actin (α-SMA) [234]. This phenotypic switching of vascular fibroblast contributes to pathological vascular remodeling by promoting fibrosis, vascular stiffening, and intimal hyperplasia [235]. Moreover, oxidative stress enhances the cross-linking of ECM proteins, further exacerbating vascular stiffness [236]. The profibrotic cytokine transforming growth factor-beta (TGF-β) is also upregulated under oxidative stress and acts synergistically with ROS to perpetuate fibroblast activation via the SMAD and non-SMAD pathways [237]. Importantly, ROS can also cause DNA damage, lipid peroxidation, and protein oxidation in fibroblasts, leading to cellular senescence, which paradoxically contributes to chronic inflammation through the senescence-associated secretory phenotype (SASP) [238]. These senescent fibroblasts remain metabolically active and secrete pro-inflammatory and pro-fibrotic mediators, sustaining a vicious cycle of oxidative stress and fibrosis [239]. Additionally, oxidative stress modulates fibroblast crosstalk with other vascular cells, including endothelial cells and smooth muscle cells. For example, ROS-induced endothelial dysfunction leads to reduced NO bioavailability and increased endothelial permeability, facilitating the infiltration of inflammatory cells and further stimulating fibroblast activation [240]. Crosstalk between fibroblasts and VSMCs under oxidative stress conditions may also contribute to VSMC phenotypic modulation and calcification [223]. Furthermore, mitochondrial dysfunction in fibroblasts under oxidative stress not only increases ROS production but also impairs cellular metabolism, shifting fibroblasts towards a glycolytic phenotype, which is often associated with fibrotic diseases [241]. In the context of vascular aging, oxidative stress speeds up fibroblast and endothelial cell senescence and contributes to age-related vascular stiffness and decreased compliance, making vessels more vulnerable to hemodynamic damage [242]. Therapeutically, targeting oxidative stress pathways in vascular fibroblasts represents a promising strategy for treating vascular diseases. In conclusion, oxidative stress is a central driver of vascular fibroblast dysfunction and contributes significantly to the pathogenesis of vascular fibrosis and remodeling. A better understanding of the redox-regulated mechanisms in fibroblasts may uncover novel therapeutic targets and improve the management of cardiovascular diseases associated with fibrosis and oxidative damage.

## 5. Role of Oxidative Stress in Vascular Diseases

### 5.1. The ROS-Aided Pathogenesis of Atherosclerosis

Oxidative stress-induced mechanisms are central to the pathogenesis of atherosclerosis, particularly in diabetes, where chronic hyperglycemia and insulin resistance enhance ROS production. Elevated ROS levels impair endothelial function by reducing NO bioavailability and increasing endothelial permeability, promoting monocyte adhesion and infiltration. ROS also oxidize LDL into oxLDL, which is taken up by macrophages to form foam cells, initiating fatty streaks. Additionally, oxidative stress activates redox-sensitive transcription factors, increasing pro-inflammatory cytokines, adhesion molecules, and matrix metalloproteinases, which drive chronic vascular inflammation, smooth muscle cell proliferation, and extracellular matrix remodeling. These processes collectively contribute to plaque formation, progression, and instability, underscoring oxidative stress as a key therapeutic target for preventing and mitigating atherosclerosis in diabetes.

There are several pathways that may contribute to the pathophysiology of oxidative stress-induced atherosclerosis (Figure 3).

#### 5.1.1. Oxidation of Lipid

During atherogenesis, LDL builds up in the arterial wall, especially in areas with disturbed blood flow [243]. This LDL is then modified by ROS produced by NOX or uncoupled eNOS [244]. Oxidized LDL promotes atherosclerosis by altering endothelial NO production and upregulating the expression of leukocyte adhesion molecules [245]. Lowering LDL oxidation in the lipoxygenases knockout mouse model has been shown to decrease atherosclerosis [246], while heightened LDL oxidation expands the lesional area [247]. Mitochondrial lipids are also significantly affected by oxidative damage from ROS [248]. Cardiolipin is the signature lipid of the mitochondrial membrane, which comprises around 20% of the total mitochondrial membrane lipid [249], and is vulnerable to oxidative damage due to its high unsaturated fatty acid content [250]. A rodent model with ischemia-reperfusion injury demonstrated reduced cardiolipin levels and elevated lipid peroxidation in liver mitochondria [251]. This can be attributed to the oxidation of accumulated succinate upon post-ischemic restoration of ETC function and subsequent elevation of mtROS produced at Complex I [252]. Elevated levels of oxidized cardiolipin and endogenous anti-cardiolipin antibodies have been observed in atherosclerotic plaques [253]. Additionally, oxidized cardiolipin acts as a pro-inflammatory signaling molecule that promotes the expression of leukotriene and 5-lipoxygenase by leukocytes, as well as the expression of adhesion molecules such as ICAM-1 and VCAM-1 by endothelial cells [254].

#### 5.1.2. Oxidation of Nucleic Acid

DNA is vulnerable to ROS-induced oxidative damage, including single-strand breaks, double-strand breaks, adduct formation, and deletions [255]. It is reported that patients suffering from CAD exhibit heightened chromosomal damage in their peripheral lymphocytes [256]. Elevated levels of 7,8-dihydro-8-oxo-2′-deoxyguanosine (8-oxo-dG), a biomarker of DNA damage in oxidative stress, are also found in plaque macrophages, smooth muscle cells, and endothelial cells [257]. Nuclear DNA damage plays an important role in the pathophysiology of atherosclerosis rather than being merely incidental. For instance, ApoE-deficient mice with haploinsufficient DNA repair enzyme kinase, ataxia telangiectasia mutated, demonstrated greater nuclear DNA damage, and accelerated atherosclerosis progression [255]. Enhanced base excision repair-induced inhibition of oxidative DNA damage significantly reduces plaque size [258]. Within mitochondria, the ROS-producing sites of Complex I are situated on the inner membrane’s matrix side, allowing superoxide to diffuse into the matrix [259]. Since mtDNA resides in the matrix and lacks protective histones, it is prone to oxidative damage from mtROS [260]. Indeed, mtDNA damage has been reported in the arteries and blood cells of patients with atherosclerosis [261,262]. Additionally, mtDNA damage worsens atherosclerosis in ApoE KO mice with impaired mtDNA polymerase proofreading activity, which correlates with signs of plaque vulnerability [261].

#### 5.1.3. Endothelial Dysfunction

Endothelial dysfunction is characterized by increased leakage, overproduction of ROS, secretion of pro-inflammatory cytokines, upregulated expression of adhesion molecules, and decreased nitric oxide (NO) production are the essential inducers of atherogenesis [263]. This dysfunctional endothelium increases the likelihood of the low-density lipoprotein (LDL) trapped in the subendothelial space and subsequent oxidative modifications [264]. The endothelium acts as both a source and target of ROS [265]. Endothelial cells predominantly produce NO due to their continuous expression of endothelial nitric oxide synthase (eNOS) [266]. Furthermore, these cells also generate superoxide and peroxynitrite because of eNOS uncoupling during tetrahydrobipterin depletion [267]. Studies have demonstrated that the absence of eNOS lowers superoxide production in ApoE-deficient mice, suggesting that eNOS uncoupling occurs during atherosclerosis [268]. Prolonged exposure to superoxide triggers mtROS production in mouse vascular endothelial cells that results in diminished proliferation [269]. Oxidative stress switches the endothelium to a pro-inflammatory state, which is essential for the pathogenesis of atherosclerosis [270]. H_2_O_2_ exposure to human endothelial cells prolonged the expression of granule membrane protein 140, which results in enhanced neutrophil adhesion to the endothelial surface [271]. Additionally, oxidative stress activates NFκB, leading to increased expression of adhesion molecules like VCAM-1, ICAM-1, and E-selectin, as well as cytokines, including TNF [272]. This TNFα stimulates the production of mtROS, activation of NOX, and expression of iNOS in endothelial cells [272,273]. Consequently, oxidative stress in the endothelium fuels inflammation, which can promote further ROS generation. Furthermore, oxidized LDL has been shown to upregulate oxidized LDL receptor-1 (LOX-1) expression in endothelial cells which results in endothelial dysfunction by inducing apoptosis and inflammation [274]. Activation of LOX-1 with oxLDL mediates the downregulation of eNOS [275]. Notably, the expression of LOX-1 in endothelial cells is correlated with the progression of atherosclerosis [276]. ApoE deficient mice with elevated endothelial LOX-1 expression exhibited increased ROS production, plaque buildup, eNOS uncoupling, and macrophage infiltration [277].

#### 5.1.4. Inflammation

ROS exerts its effects by activating various targets, including NFĸB, HIF-1α, and the NLRP3 inflammasome, to drive inflammation, a critical factor in atherogenesis [278]. NFĸB serves as a pivotal transcription factor that regulates the transcription of a large number of pro-inflammatory cytokines, including TNFα, IL-1β, and IL-18 [278,279]. In HUVECS, hydrogen peroxide treatment activates NFĸB through the tyrosine phosphorylation of IKB, the upstream target of NFĸB activation [280]. Additionally, superoxide generated by NOX can also activate NFĸB [281]. Lipopolysaccharide (LPS), a component of bacterial cell walls, binds to TLR4, which partners with NOX4 to promote ROS production and subsequent activation of NFĸB in HEK293T cells [282]. Endothelial NOX activation occurs under low shear stress conditions and subsequently enhancing NFĸB signaling in atherogenesis [59,283]. Bone morphogenic protein 4 is implicated to upregulate the expression of NOX1 mRNA following the production of H_2_O_2_ and superoxide [59]. Furthermore, HIF-1α has been shown to mediate ROS-induced expression of inflammatory cytokine [284]. Upon LPS stimulation, monocytes undergo a significant metabolic shift to differentiate into M1 macrophages, which depend on glycolysis for ATP production. This metabolic shift results in elevated glucose uptake and decreased reliance on mitochondrial oxidative phosphorylation (OXPHOS), both necessary for the expression of certain pro-inflammatory factors, such as IL-1β [285]. The diminished reliance on OXPHOS results in succinate accumulation, which can be oxidized by succinate dehydrogenase, spurring ROS production and stabilizing HIF-1α to boost IL-1β expression [286]. ROS also contribute to inflammation via the NLRP3 inflammasome [287]. The activation of the NLRP3 inflammasome is crucial for generating mature IL-1β and IL-18 [288]. The NLRP3 inflammasome consists of NLRP3, the ASC adapter protein, and caspase 1 that processes pro-IL-1β and pro-IL-18 into their active forms [289]. ROS derived from xanthine oxidase are pivotal for regulating the activation of NLRP3 inflammasome in macrophages [290]. ROS-induced NLRP3 activation increases IL-1β and IL-18 expression, whereas xanthine oxidase inhibition with febuxostat attenuated this effect [291]. Mitochondrial oxidative stress also plays a significant role in regulating the activity of NLRP3 inflammasome [292]. Increasing mtROS by inhibiting ETC enzymes or disrupting autophagy/mitophagy induces NLRP3 activation [293]. As a damage-associated molecular pattern, MtROS-induced oxidation of mtDNA activates the NLRP3 inflammasome [294]. Thus, ROS can enhance both the transcription and post-translational modifications of inflammatory cytokines. IL-1 expression is critical in atherosclerosis. IL-1β plays a crucial role in atherogenesis as it promotes the expression of adhesion molecules and chemokines, including ICAM-1, VCAM-1, and MCP-1, which recruit leukocytes and mononuclear phagocytes during the early stages of atherosclerosis development [295,296]. Furthermore, IL-1β acts as a potent mitogen for human SMC and induces further inflammation by auto-induction and IL-6 stimulation [297,298]. The significance of IL-1 in atherosclerosis was demonstrated in the CANTOS trial, where treatment with canakinumab, a monoclonal antibody targeting IL-1, reduced the primary endpoint of non-fatal myocardial infarction, non-fatal stroke, or cardiovascular death [299].

#### 5.1.5. Destabilization of Fibrous Cap

Atherosclerotic plaque rupture results in thrombus formation that blocks blood vessels, leading to stroke and myocardial infarction [300]. This plaque rupture occurs due to the degradation of the fibrous cap that protects the plaque [301]. Destabilization of the plaques is determined by the high ratio of macrophages to VSMCs, a substantial lipid-rich necrotic core, and a thin fibrous cap [302]. The formation of fibrous caps involves the deposition of collagen-rich extracellular matrix (ECM) components along with the buildup of VSMCs [303]. Matrix metalloproteinases (MMPs) are pivotal in every phase of atherogenesis by regulating vascular inflammation, endothelial dysfunction, VSMCs migration, vascular calcification, extracellular matrix breakdown, as well as plaque activation and destabilization [304]. MMPs exert differential effects on plaque structure and development, potentially depending on the disease stage. For instance, upregulation of MMP-1, MMP-8, and MMP-13 is implicated in atherosclerotic plaque development and ECM collagen degradation [305]. Additionally, the upregulation of gelatinases MMP-2 and MMP-9 that cleave collagen is implicated in destabilized plaques [306,307]. Consistent with these findings, MMP-9 expressed in macrophages leads to plaque disruption, while MMP-12/ApoE double knockout mice exhibit increased VSMC content in plaques [308,309]. Contrary to the previous findings, MMP-3/ApoE double knockout mice showed larger plaque area, implying a protective function [309]. The expression and activity of MMPs are affected by ROS. OxLDL has been shown to upregulate the expression of MMP-1, MMP-2, and MMP-9 [310,311,312], whereas H_2_O_2_ activates MMP-1 and MMP-2 [313]. Superoxide derived from monocyte NOX correlates with plasma levels of MMP-9, enhances MMP-9 activity, and is linked to reduced plaque collagen content [314]. Therefore, the above studies suggest that vascular ROS promotes plaque stability by modulating the expression and activities of MMPs.

### 5.2. The Role of Oxidative Stress in the Pathophysiology of DR

DR is classified as a microvascular complication of diabetes, which poses significant risks to vision [315]. Approximately 400 million individuals globally are diagnosed with type 2 diabetes, with more than 45% of this population experiencing DR [316]. It is widely recognized that DR constitutes the primary cause of diabetes-related visual impairment or loss among working-age adults and the elderly across the globe [317]. Projections indicate that the number of patients suffering from DR may increase to 191.0 million by the year 2030 [318,319,320]. The primary pathophysiology of DR encompasses various alterations induced by hyperglycemia, which include the thickening of the retinal capillary basement membrane, elevated retinal vascular permeability, retinal tissue ischemia, and the upregulation of multiple vasoactive substances, resulting in neovascularization [321]. There are two major types of DR, including nonproliferative (mild, moderate or severe), and proliferative DR [322]. Non-proliferative DR is characterized by the absence of neovascularization, formation of microaneurysms and mild dilation of retinal blood vessels, which is associated with the earliest stages of DR progression [323,324], whereas proliferative DKD is characterized by the formation of new blood vessels on the retinal surface [325]. Due to the inherent instability of these newly formed blood vessels in proliferative DR, the constituents within the vessels, such as blood and extracellular fluids, are prone to leakage, leading to complications such as vitreous hemorrhage and retinal detachment that culminate in vision loss [326].

Oxidative stress is a key factor in the progression of DR. The excessive buildup of ROS damages the tissues surrounding the retinal microvasculature, ultimately contributing to the onset of DR [327]. Studies suggested four principal metabolic abnormalities that contribute to the pathophysiology of hyperglycemia-induced oxidative damage in the retina: (1) activation of the PKC pathway, (2) flux through the polyol pathway, (3) activation of the hexosamine pathway, and (4) intracellular formation of AGEs [328,329]. In addition to these metabolic issues, irregular epigenetic modifications [330], impaired activity of nuclear factors such as the upregulation of NFκB (nuclear factor κB) [331,332] and the downregulation of nuclear factor erythroid 2-related factor 2 (Nrf2) activities [333,334], and mitochondrial dysfunction [335] have also been implicated in the overproduction of ROS in DR. Importantly, oxidative stress driven by epigenetic changes can persist over time, even after blood glucose levels normalized [336]. This phenomenon is known as “metabolic memory [336].” Moreover, hyperglycemia-induced oxidative stress contributes to apoptosis of retinal cells, inflammation, lipid peroxidation, and notable structural and functional changes, including microvascular complications and neurodegeneration in the eyes associated with DR [337]. There are several pathways that may contribute to the pathophysiology of oxidative stress-induced DR (Figure 4).

#### 5.2.1. Mitochondrial Dysfunction

In hyperglycemia, oxidative stress-induced elevated ROS generation leads to mitochondrial dysfunction. The displacement-loop (D-loop) in mitochondrial DNA comprises a substantial non-coding sequence and is a highly vulnerable unwound region, containing essential elements for transcription as well as control regions for mitochondrial DNA (mtDNA) replication [338,339]. In DM, the D-loop experiences more significant impairments and mutations compared to other sections of mtDNA, leading to a reduction in its copy numbers. Furthermore, in DR, hyperglycemia-induced hypermethylation of mtDNA adversely affects its transcription, culminating in mitochondrial dysfunction, which ultimately promotes the apoptosis of retinal microvascular cells [340]. Epigenetic modification of mtDNA has also been corroborated as a latent factor contributing to base mismatch in mtDNA during the pathogenesis of DR [341]. mtDNA-encoded proteins are crucial for the normal functioning of the ETC and mitochondrial homeostasis [342]. In contrast to nuclear DNA, circular mtDNA is susceptible to oxidative stress-induced extensive and persistent damage, due to the absence of protective histones [343]. Damaged mtDNA compromises transcription and protein synthesis, further undermining electron transport and exacerbating ROS generation [340]. Additionally, the activation of matrix metalloproteinases (MMPs) is implicated in mitochondrial dysfunction in DR [344]. Oxidative stress in diabetes upregulates MMPs expression [345] through the activation of the Nox complex [346]. Oxidative stress in diabetes has been shown to increase the translocation of MMPs into the mitochondria [347]. This process involves the translocation and accumulation of redox-sensitive MMPs, specifically MMP-2 and MMP-9, within the retinal mitochondria. This translocation depends on the modulation of chaperones Hsp60 and Hsp70 [348,349]. Once inside the mitochondria, MMPs disrupt mitochondrial function and increase pore permeability by damaging connexin 43 [350,351]. The compromised lipid membrane of the mitochondria leads to mitochondrial swelling in the diabetic retinas [352] and promotes the leakage of cytochrome c (Cyt c) into the cytosol, which in turn triggers the assembly of the apoptosome platform to initiate caspase cascade [353,354]. Additionally, peroxynitrite, an extremely reactive molecule, is formed when superoxide reacts with nitric oxide (NO) [355]. Peroxynitrite oxidizes glutathione (GSH), cysteine, and tetrahydrobiopterin [356], and subsequently oxidizes membrane phospholipids, inactivates enzymes containing sulfhydryl groups, triggers the nitration of tyrosine residues, and exacerbates DNA fragmentation [357]. Furthermore, peroxynitrite causes irreversible damage to mitochondria by disrupting mitochondrial energy and calcium homeostasis and promoting the opening of the permeability transition pore, ultimately leading to cell apoptosis [358].

#### 5.2.2. Cellular Apoptosis and Inflammation in the Retina

In DR, the apoptosis of retinal cells occurs early on. The accelerated death of retinal capillary cells is evident before any histopathological changes linked to this complication appear [214,359]. Hyperglycemia-induced oxidative stress in retinal endothelial cells and pericytes upregulates the activities of Caspase-3, NFκB, and other transcription factors that promote capillary cell apoptosis [329,360]. Caspases play an important role in apoptosis and are vulnerable to oxidative stress-induced damage [361,362]. Hyperglycemia-induced ROS generation in the mitochondria increases the organelle’s pore permeability [363,364], leading to the release of Cyt c and other pro-apoptotic factors to trigger apoptosis by activating caspases [365,366]. At the early stage in apoptosis, Cyt c in retinal capillary cells [367] activates Caspase-9 and subsequently activates Caspase-3, leading to DNA fragmentation [367,368]. Furthermore, NFκB plays an important role in hyperglycemia-induced inflammation and cell death due to its proinflammatory and proapoptotic properties [369,370]. The activity of retinal NFκB in diabetes is upregulated early in retinopathy development and remains active even as apoptosis in retinal capillary cells accelerates [371]. As a redox-sensitive nuclear transcription factor, NFκB significantly regulates inflammatory responses and inhibits antioxidant enzymes [372]. NFκB is implicated in starting proapoptotic processes in retinal pericytes in response to high glucose, explaining the early pericyte death in DR [373]. Upregulation of NFκB signaling pathways can activate MMP-9 [374], leading to increased mitochondrial MMP-9, which damages the mitochondrial gap junction protein connexin-43 and further enhances pore permeability. This contributes to the leakage of Cyt c into the cytosol [351]. Additionally, NFκB increases NO production by upregulating the expression of inducible nitric oxide synthase [375,376], which upregulates NFκB’ s transcriptional activity [377].

#### 5.2.3. Lipid Peroxidation

Oxygen-derived free radicals, including hydroxyl and hydroperoxyl species, have been shown to oxidize phospholipids and other lipid components in the plasma membrane, resulting in lipid peroxidation. Tissues affected by oxidative stress-related retinal diseases have demonstrated a significant presence of metabolic products from lipid peroxidation, thereby indicating a correlation between oxidative stress and lipid peroxidation [378]. Hyperglycemia-induced oxidative stress in DM plays a pivotal role in augmenting lipid peroxidation in DR [379]. Furthermore, lipid peroxidation exhibits a positive association with the severity and duration of DM [380]. The high concentration of polyunsaturated fatty acids (PUFAs) in the retina enhances its susceptibility to oxidative stress due to the sensitivity of PUFAs to oxidation [381]. Intracameral administration of H_2_O_2_ into the rabbit retina resulted in an increase in lipid peroxidation within the membranes of iris epithelial cells [382]. Additionally, lipid metabolism contributes to the generation of ROS, such as H_2_O_2_, which facilitates the senescence of retinal pigment epithelial (RPE) cells, thereby exacerbating the progression of DR. Concurrently, lipid peroxidation products may promote the leakage of ROS from mitochondria [383]. Lipid peroxidation results in cellular membranes with compromised integrity and the formation of diffusible cytotoxic reactive aldehydes, including 4-hydroxy-2-nonenal (4HNE) and 4-hydroxyhexenal (HHE). 4HNE has been shown to activate the canonical WNT signaling pathway through oxidative stress that leads to the pathogenesis of DR [381]. Additionally, 4HNE induces apoptosis of RPE cells through p53 activation [384]. HHE has been shown to induce permeable transition pores in the mitochondrial membrane and augment NFκB-induced upregulation of pro-inflammatory genes [385]. Research on neurodegenerative models of the retina has shown an increase in lipid peroxidation that correlates with neuronal loss [386]. Hyperglycemia-induced ROS generation in mitochondria directly contributes to mitochondrial dysfunction. Consequently, ROS-mediated mitochondrial dysfunction results in the accumulation of lipid droplets within glial cells that may be peroxidized by ROS, leading to the initiation of neurodegeneration in the retina [387].

#### 5.2.4. Changes in Retinal Microvasculature

Oxidative stress-induced metabolic impairments lead to various effects on the retinal microvasculature’s structure and function, including the thickening of the capillary basement membrane (CBM), disruption of the blood-retinal barrier (BRB), and the formation of acellular and occluded capillaries [388]. The thickening of the CBM is a consistent characteristic of DR, due to upregulated expression and decreased degeneration of extracellular matrix (ECM) proteins [389]. Hyperglycemia-induced oxidative stress and advanced glycation in diabetes contribute to the thickening of CBM in DR [390]. Hyperglycemia-induced ROS generation increases the activity of proinflammatory transcription factors that lead to thickening of the CBMs by upregulating the expression of ECM proteins, including fibronectin (FN) and collagen, in retinal endothelial cells [389,391]. AGE formation from ROS on collagen causes cross-linking, resulting in structural rigidity and limiting the transport of growth factors across membranes, which contributes to the loss of pericytes and endothelial cells [392]. Treatment with the AGE inhibitor aminoguanidine has shown protective effects against CBM thickening in diabetic rats [393]. Structural changes induced by oxidative stress, such as the loss of intercellular junctions and pericyte apoptosis, correlate with functional changes like alterations in blood flow and increased vessel permeability, both of which contribute to the pathogenesis of DR [394]. Oxidative stress is implicated in the damage of the BRB, a selective barrier that regulates substance exchange between the neural retina and circulating blood, ensuring retinal health by supplying nutrients and removing metabolic waste and toxins [395]. BRB breakdown leads to increased vascular permeability, leading to diabetic macular edema, a significant cause of vision loss among diabetic patients. ROS-induced elevation of VEGF levels plays a key role in BRB destruction [395], confirmed through intravitreal VEGF injections or implantation of VEGF-releasing pellets [396,397]. Occludin, claudin, and zonula occludens-1 (ZO-1) are intercellular junction proteins, essential for maintaining BRB integrity, and the breakdown of the BRB is closely linked to the dysfunction of these junction-associated proteins. For instance, oxidative stress upregulates VEGF expression, which in turn induces the expression of urokinase plasminogen activator receptor (uPAR), which results in the breakdown of the BRB. Inhibiting VEGF or uPAR expressions has been found to preserve BRB integrity in diabetic models [398]. Prolonged exposure to VEGF results in the loss of Claudin-1, jeopardizing BRB integrity [396]. Furthermore, oxidative stress-induced activation of NFκB in DR, regulates the transcription of various genes, including suppression of ZO-1 by this inflammatory regulator, which compromises BRB integrity [399]. Chronic hyperglycemia-induced activation of PKC-δ, initiating a signaling cascade that results in pericyte apoptosis [400]. As pericytes die, the structure of retinal microvessels is gradually altered, leading to BRB disruption [316]. Secretion of proangiogenic cytokines, growth factors, and proteases by proinflammatory cells has been shown to promote angiogenesis at inflammation sites in DR [401]. Oxidative stress-induced upregulation of prostaglandin E2 (PGE2) and Cyclooxygenase-2 (COX-2), promotes the expression of proangiogenic factor, VEGF, in diabetic retinas, which increases retinal neovascularization [402,403,404]. Oxidative stress-induced apoptosis of retinal neurons and pericytes in DR augments the growth of acellular and occluded capillaries, leading to microaneurysm formation and increased leukostasis, which contributes to CBM thickening [316]. Endothelin-1 is an important vasoactive peptide that regulates blood flow and permeability in the retinal microvasculature. ROS upregulates the transcription of endothelin-1 [405,406], which can promote CBM thickening by increasing the synthesis of extracellular matrix proteins [407,408].

### 5.3. The Role of Oxidative Stress in the Pathophysiology of DKD

DKD is attributed to complex interactions among various pathways, initiated by hyperglycemia and hemodynamic changes associated with diabetes [409]. These changes lead to albuminuria, deteriorating kidney function, renal fibrosis, and inflammation. Additionally, oxidative stress is recognized as a key factor connecting hyperglycemia to vascular complications of diabetes, especially DKD [410]. There are several pathways that may contribute to the pathophysiology of oxidative stress-induced DKD (Figure 5).

#### 5.3.1. Pi3k/Akt Signaling Pathway

The phosphoinositide 3-kinase (PI3K)/protein kinase B (Akt) signaling pathway regulates cell proliferation, differentiation, and apoptosis. In response to extracellular stimuli, PI3K phosphorylates phosphatidylinositol 4,5-bisphosphate (PIP2), leading to the production of phosphatidylinositol-3,4,5-triphosphate (PIP3). The resulting PIP3 subsequently induces the translocation of Akt to the plasma membrane and activates Akt through the enzymatic action of phosphoinositide-dependent kinase-1 (PDK1) [411]. The activated Akt then in turn activates the downstream signaling molecules, including glycogen synthase kinase (GSK) 3β, mammalian target of rapamycin (mTOR), and forkhead box protein O1 (FoxO1) to show its pathophysiological effects [412].

GSK3β is one of the downstream targets of Akt. Activated Akt antagonizes the pathophysiological effects of GSK3β by phosphorylating it. Studies demonstrated that chronic high glucose exposure suppresses PI3K/Akt signaling pathway, resulting in decreased GSK3 activity, reduced Bcl-2, and increased Bax, which induces renal cell apoptosis [413]. An in vivo study in DKD mice demonstrated that phosphorylation of GSK3β increased inflammation by augmenting the activity of NFκB [414]. Another study indicated that diminished GSK3β activity elevated β-catenin levels and slowed the degradation of Snail, which enhanced the trans-differentiation of tubular epithelial cells and fostered renal interstitial fibrosis in diabetic rats [415].

mTOR has emerged as a key research target because of its strong association with Akt-mediated signaling pathways. When Akt is activated, it phosphorylates mTOR and its downstream pathways, affecting cellular functions such as proliferation, apoptosis, and glucose metabolism [416]. Elevated glucose levels have been shown to promote tubular epithelial NRK-52E cells to produce ROS, which in turn trigger the release of transforming growth factor-beta 1 (TGF-β1) and activated Akt. The activated Akt phosphorylates mTOR, facilitating the epithelial-mesenchymal transition (EMT) in NRK-52E cells, exacerbating diabetic kidney fibrosis [417]. Furthermore, high glucose levels are implicated in stimulating mTOR signaling, leading to a decrease in the LC3II/LC3I ratio and reduced Beclin levels, which inhibit autophagy and induce the progression of DKD [418].

FoxO1 consists of a set of highly conserved transcription factors found throughout the body, playing vital roles in oxidative stress, inflammation, autophagy, and apoptosis in response to elevated glucose levels [419]. Phosphorylation of FoxO1 by the PI3K/Akt signaling pathway results in its inactivation of nuclear translocation, which is essential for the onset of diabetic kidney injury [420]. In renal tissues, FoxO1 can aid in repairing the damaged glomerular filtration barrier and reducing apoptosis caused by the detachment of renal podocytes from the basement membrane due to impaired glucose metabolism [421]. An initial spike in glucose levels activates the TGF-β/Smad pathway, which helps mitigate kidney fibrosis in DKD [422]. Additionally, elevated phosphorylation of FoxO1 downregulated its activity and autophagy in DKD animal models, aggravating kidney damage [423].

#### 5.3.2. NFκB Signaling Pathway

Under normal physiological conditions, NFκB is expressed throughout various cells in an inactive form. When stimulated by agonists, including proinflammatory cytokines, Toll-like receptors, p38-MAPK, HO-1, and ROS, NFκB dimers (p65 and p50) are translocated into the nucleus to regulate the expression of target genes, responsible for the body’s immune and inflammatory responses [424]. It is already well studied that upregulation of NFκB activity is linked to the pathophysiology of DKD. For instance, hyperglycemia-induced generation of ROS and inflammatory factors, including TNF and IL-6 activate the NFκB signaling pathway, and subsequently upregulates the transcription of proinflammatory cytokines, chemokines, adhesion molecules, and TGF-β1, which ultimately leads to cellular apoptosis, necrosis, and tissue fibrosis, all of which expedite the progression of DKD [425]. A study demonstrated that NFκB expression was significantly elevated in DKD patients compared to non-diabetic subjects, and this expression level correlated positively with the severity of proteinuria in those affected [426]. All these studies collectively suggest that, attenuating the upregulation of NFκB signaling pathway may offer a promising avenue for research aimed at preventing and treating DKD.

#### 5.3.3. Nrf2/ARE Signaling Pathway

Nuclear factor erythroid 2-related factor 2 (Nrf2) is a crucial transcription factor that enhances the body’s defense against oxidative stress. It regulates the transcription of several antioxidants, anti-inflammatory, anti-apoptotic, and anti-fibrotic factors, and its activity is negatively correlated with the severity of DKD. Under normal physiological circumstances, Nrf2 is found in the cytoplasm in an inactive form conjugated with its inhibitor, Kelch-like ECH-associated protein 1 (KEAP1). However, hyperglycemia-induced oxidative stress rescues Nrf2 from KEAP1, leading to its activation. Once activated, Nrf2 translocates into the nucleus and interacts with AREs to upregulate the expression of antioxidant factors, thereby protecting cells and alleviating the progression of kidney injury in DKD [427]. A study in mice revealed that *Nrf2^−/−^* mice experienced more severe kidney damage compared to wild-type control mice, suggesting the protective role of Nrf2 in kidney diseases [428]. Another study demonstrated that inhibiting Nrf2-mediated antioxidant signaling pathway downregulated the transcription of antioxidant enzymes, such as SOD, catalase (CAT), and glutathione peroxidase (GSH-Px), and ultimately worsened kidney injury in mice with spontaneous DKD [429]. Additionally, a study in rodents found that blocking Nrf2 signaling while activating TGFβ-smad signaling resulted in a delay in renal fibrosis in STZ-induced diabetic rats [430]. Furthermore, increasing Nrf2 levels are associated with a reduction in TNF, IL-6, Bax, and p38 levels, suppressed NFκB activation, and upregulated Bcl-2 expression, demonstrating anti-inflammatory and anti-apoptotic effects in STZ-induced DKD rats [431]. All these studies collectively suggest that Nrf2 could serve as a critical target for preventing DKD and preserving renal function.

#### 5.3.4. TGFβ Signaling Pathway

Oxidative stress-induced activation of the TGF-β1 signaling pathway upregulates the expression of protein kinases or cytokines, followed by ECM accumulation and EMT, ultimately resulting in renal interstitial fibrosis and glomerulosclerosis in DKD [432]. A study in a human cohort indicated that serum TGF-β1 levels can be served as a potential biomarker for early detection of fibrosis in DKD [433]. Resveratrol, Salvia Root, Taxol, and Calcitriol are potential drugs that alleviate fibrosis in DKD by antagonizing the TGF-β1 signaling pathway [434]. One of the major downstream targets of TGF-β1-induced profibrotic activities in the pathogenesis of DKD is ERK. The activation of the TGF-β1/ERK pathway is implicated in the pathogenesis of kidney fibrosis. Multiple studies in animal model demonstrated an improvement in kidney fibrosis upon inhibition of the TGF-β1/ERK pathway [435].

#### 5.3.5. JAK2/STAT3 Signaling Pathway

DKD is associated with an augmented activation of Janus kinase (JAK)/signal transducer and activator of transcription (STAT) signaling pathway, whereas inhibiting this pathway ameliorate DKD progression [436]. A study in DKD rats showed elevated phosphorylation of JAK and STAT3 in renal tissue, which results in an upregulation of Bax, and a downregulation of Bcl-2 expression followed by an increased renal cell apoptosis [437]. Another study in mouse with diabetic nephropathy demonstrated that the activation of JAK2/STAT3 signaling pathway in the kidney residual macrophages induced the release of proinflammatory factors and ROS, worsening kidney damage [438]. CXCL6 may stimulate the production of TGF-β1, collagen I, collage III, MMP2, and MMP9 through the activation JAK/STAT pathway, and thereby accelerating the progress of renal fibrosis [439]. These findings collectively suggest that the JAK2/STAT3 signaling pathway primarily plays a role in governing immune response, inflammation, oxidative stress, and cell apoptosis throughout the development of DKD.

#### 5.3.6. AMPK Signaling Pathway

AMPK is well-known as a cellular energy sensor and has garnered significant attention as a therapeutic target for obesity, diabetes, and diseases related to metabolic syndrome. Studies in humans demonstrated that the expression of phosphorylated AMPK (p-AMPK) protein levels is lower in the renal tissues of DKD patients. Furthermore, treatment with an AMPK agonist has been found to alleviate clinical symptoms, improve pathological changes, and reduce 24-h urinary protein levels in these patients [440]. Another study in diabetic rats showed that inactivation of AMPK in the renal cortex leads to NOX4 up-regulation, activation of TGFβ-1 signaling and increased ECM accumulation [441]. Upon activation, AMPK has been shown to increase the expression of SODs to diminish superoxide production and phosphorylates and activates FOXO to provide antioxidant effects [442]. Both in vivo and in vitro studies showed that AMPK, p-AMPK, SIRT1, NFκB levels were significantly reduced in STZ-induced DKD mice and in high-glucose-treated podocytes, triggering oxidative stress-induced pyroptosis [443]. High glucose-induced inhibition of AMPK phosphorylation upregulates the expression of PPAR-α, pro-inflammatory and profibrotic genes in proximal tubular epithelial cells (PTECs), promotes intrarenal lipid accumulation and apoptosis of PTECs [444]. Another study demonstrated that agonist-stimulated activation of AMPK increases mitophagy through the p-AMPK/Pink1/Parkin signaling pathway, thereby ameliorate renal oxidative stress and tubulointerstitial fibrosis in high-fat diet/STZ-induced diabetic mice [445].

## 6. Therapeutic Approaches to Tackle Oxidative Stress-Induced DVDs

Oxidative stress remains a central pathological mechanism underlying vascular complications in diabetes, driven primarily by persistent hyperglycemia, dyslipidemia, and insulin resistance, which collectively promote excessive production of ROS through mitochondrial dysfunction, NOX activation, and eNOS uncoupling. These pathways lead to endothelial dysfunction, vascular inflammation, and eventual macro- and microvascular complications [6]. As a result, therapeutic strategies have progressively shifted from conventional antioxidants to mechanistically targeted interventions. Among glucose-lowering agents, sodium glucose co-transporter 2 (SGLT2) inhibitors, including dapagliflozin, empagliflozin, and canagliflozin have demonstrated pleiotropic vascular protective effects beyond glycemic control. These agents mitigate mitochondrial ROS production, suppress NOX4 expression, activate the AMPK/SIRT1/PGC-1α axis, preserve mitochondrial membrane potential, and enhance eNOS phosphorylation, culminating in improved endothelial function and reduced arterial stiffness in both preclinical and human studies [446]. Similarly, thiazolidinediones (TZDs), such as pioglitazone and rosiglitazone, exert antioxidant effects by activating peroxisome proliferator-activated receptor gamma (PPARγ), reducing systemic and vascular oxidative stress, and attenuating NOX-mediated ROS production, partly through AMPK activation [447,448]. Another major class of therapeutic agents showing promise is glucagon-like peptide-1 receptor agonists (GLP-1 RAs), including liraglutide, semaglutide, and dulaglutide. These drugs reduce ROS generation through multiple mechanisms: by enhancing mitochondrial biogenesis and fusion (via PGC-1α), activating cAMP/PKA and PI3K/Akt pathways, increasing expression of antioxidant enzymes such as SOD2 and catalase, and restoring endothelial NO bioavailability [449,450]. Recent studies also demonstrate that GLP-1 RAs inhibit NOX activity and reduce inflammatory cytokine production in endothelial and vascular smooth muscle cells [451,452]. Importantly, large clinical trials like LEADER and SUSTAIN-6 have established cardiovascular benefits of GLP-1 RAs, correlating with improved vascular oxidative s tress biomarkers in patients with type 2 diabetes [453].

Beyond glucose-lowering agents, therapeutic approaches increasingly target endogenous antioxidant defense systems. Nrf2 activation via phytochemicals like sulforaphane or pharmacological agents such as bardoxolone methyl enhances transcription of AREs including heme oxygenase-1 (HO-1), NAD(P)H quinone oxidoreductase-1 (NQO1), and glutathione peroxidase (GPx), thereby reducing vascular inflammation and oxidative injury [454,455]. Although bardoxolone methyl initially raised concerns regarding fluid retention in the BEACON trial, refined Nrf2 modulators (e.g., **omaveloxolone**) are currently in clinical evaluation with improved safety profiles [456]. NOX inhibitors have emerged as another promising avenue. For instance, GKT137831 selectively inhibits NOX1 and NOX4, reducing ROS generation and fibrotic signaling in diabetic vasculature [457]. These findings were corroborated in animal models of diabetic atherosclerosis and nephropathy, although human trials remain limited. Mitochondria-targeted antioxidants such as MitoQ, SS-31 (elamipretide), and MitoTEMPO represent advanced therapeutic tools capable of accumulating within the mitochondrial matrix to neutralize mtROS at the source. These agents restore mitochondrial membrane potential, suppress lipid peroxidation, and prevent mitochondrial DNA damage, ultimately preserving vascular endothelial function [458,459]. For instance, SS-31 has shown efficacy in reducing vascular stiffness and improving endothelial-dependent vasodilation in older adults with impaired mitochondrial energetics [460].

A novel frontier in redox biology is the regulation of ferroptosis, an iron-dependent, lipid peroxidation-driven form of regulated cell death which has been implicated in the progression of diabetic vasculopathy, nephropathy, and cardiomyopathy. High glucose environments promote ferroptosis by suppressing GPx4 activity and depleting glutathione, leading to accumulation of oxidized phospholipids and endothelial cell death [461]. Interventions such as ferrostatin-1, deferoxamine, or curcumin have shown protective effects against ferroptosis in vascular tissues under hyperglycemic conditions [462,463]. Nano-formulated curcumin exhibits enhanced bioavailability and ROS scavenging properties, suppressing ferroptotic gene expression and improving vascular remodeling in diabetic models [464,465,466]. In parallel, emerging nanomedicine platforms allow for targeted, ROS-responsive delivery of antioxidants or gene-editing cargo. Nanocarriers such as polymeric nanoparticles, lipid vesicles, or exosomes can deliver antioxidants like resveratrol, mangiferin, and metformin directly to sites of vascular inflammation, enhancing therapeutic efficacy while minimizing systemic toxicity [467,468,469].

Gene-based interventions are also being explored, including the delivery of plasmids or viral vectors encoding antioxidant enzymes (e.g., SOD2, catalase, Nrf2) directly to vascular tissues, or CRISPR-based activation of protective genes [470,471,472,473]. These strategies aim to restore redox balance in a durable, cell-type-specific manner. Additional therapeutic targets include epigenetic regulators of redox homeostasis. Histone deacetylase (HDAC) inhibitors and SIRT1 activators (such as resveratrol or nicotinamide mononucleotide) have demonstrated benefits in improving insulin sensitivity, suppressing inflammatory gene transcription, and enhancing mitochondrial antioxidant defenses [474,475,476]. Modulation of the AGE pathway via RAGE antagonists or pro-resolving mediators like Annexin A1 further reduces vascular ROS generation and leukocyte adhesion in diabetic endothelium [477,478]. Adjunct lifestyle interventions such as caloric restriction, time-restricted feeding, and Mediterranean diets rich in flavonoids also support vascular redox homeostasis by activating AMPK, Nrf2, and sirtuin pathways, while aerobic exercise upregulates endogenous antioxidant enzymes and improves mitochondrial function [479]. Importantly, combination therapies that integrate metabolic agents (e.g., SGLT2 inhibitors, GLP-1 RAs) with redox modulators (e.g., NOX inhibitors, ferroptosis blockers, or Nrf2 activators) show additive or synergistic effects in diabetic animal models and are being increasingly explored in early-phase clinical trials [480,481,482,483]. The pathogenic model of most DVDs centers heavily on oxidative stress as a major contributor. ROS, such as superoxide and hydrogen peroxide, mediate oxidative stress-induced endothelial dysfunction, oxLDL formation, vessel wall inflammation, and smooth muscle proliferation. *In vitro* and *in vivo* studies consistently support this model. Antioxidants ranging from vitamins C and E to flavonoids, coenzyme Q10, and mineral cofactors effectively quench ROS, enhance nitric oxide availability, and preserve endothelial function in experimental settings. These findings underpin the rationale for antioxidant therapies in human vascular disease [484].

However, this elegant mechanistic narrative starkly contrasts with the results of large-scale human trials. Multiple high-powered randomized clinical trials (RCTs) testing antioxidant vitamins in vascular disease have failed to demonstrate benefit and, in some cases, suggested harm. For instance, vitamins E, C, D, β-carotene showed no significant reduction in cardiovascular events in major secondary prevention trials [484,485]. Succinobucol, a potent lipoprotein-associated antioxidant did not reduce/improve events in the ARISE trial on 6144 patients with acute coronary syndrome [485]. The HOPE-TOO trial evaluated whether long-term (over 7 years) supplementation with vitamin E would reduce cardiovascular events in high-risk cardiovascular patients (n ≈ 7030). However, no significant reduction in major cardiovascular events, including myocardial infarction, stroke, and cardiovascular death was observed in the vitamin E group compared to placebo [486]. Coenzyme Q10 (CoQ10) [296], resveratrol [486], melatonin [487], N-Acetylcysteine (NAC) [488], mitochondria-targeted antioxidant (mitoQ) [489,490] in various hypertensive and cardiovascular trials showed occasional biomarker improvements (e.g., blood pressure, flow-mediated dilation) but no consistent clinical benefit. The failure of potential antioxidant therapy is stated in Table 2.

It is now well appreciated that the failure of large-scale antioxidant trials in the past such as the HOPE-TOO trial, which showed no cardiovascular benefit from vitamin E and C in high-risk patients was largely due to their non-specificity, failure to distinguish between physiological and pathological ROS, and lack of targeted delivery [495]. In contrast, the current precision-based strategies emphasize spatial and temporal control over oxidative stress, employing molecular diagnostics and biomarkers (e.g., oxidized LDL, 8-iso-PGF2α, or MDA levels) to guide treatment. This evolving therapeutic paradigm seeks not to eliminate ROS entirely, but rather to recalibrate redox signaling pathways that are aberrantly activated in diabetes. As our understanding of redox systems deepens, and as tools for targeted delivery and pathway-specific modulation continue to advance, there is renewed optimism that oxidative stress-induced vascular disease in diabetes may become more tractable. Future directions should focus on biomarker-guided clinical trials, combination regimens tailored to specific vascular phenotypes, and the integration of redox therapeutics with other emerging technologies such as single-cell transcriptomics and vascular imaging to assess real-time ROS dynamics. Together, these developments mark a significant leap forward in the quest to combat diabetes-associated vascular disease through redox-based precision medicine.

## 7. Conclusions

In conclusion, diabetes-induced oxidative stress plays a pivotal role in the pathogenesis of vascular diseases, contributing significantly to morbidity and mortality in diabetic populations. Chronic hyperglycemia drives excessive ROS production, leading to lipid, protein, and DNA damage and compromising vascular integrity. This oxidative environment induces endothelial dysfunction by reducing NO bioavailability, promoting vasoconstriction, inflammation, and thrombosis. Activation of redox-sensitive transcription factors further amplifies inflammation by upregulating adhesion molecules and cytokines, facilitating leukocyte infiltration into the endothelium and accelerating vascular complications in the heart, kidneys, and eyes. Emerging evidence suggests that targeting oxidative stress through antioxidant therapies, lifestyle interventions, and pharmacological agents may help preserve vascular function in diabetes. However, clinical translation remains challenging, underscoring the need for precise, personalized strategies to modulate redox balance. Addressing oxidative stress is essential for preventing vascular complications and improving outcomes in diabetic patients.

## Figures and Tables

**Figure 1 medsci-13-00087-f001:**
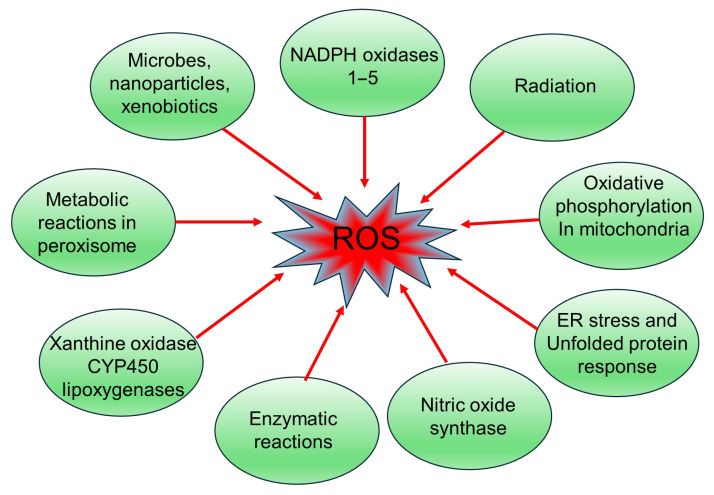
Sources of ROS. ROS production arises from NADPH oxidases (isoforms 1–5), exposure to radiation, oxidative phosphorylation within mitochondria, and endoplasmic reticulum (ER) stress with unfolded protein response. Additional sources include nitric oxide synthase activity, enzymatic reactions, and oxidative enzymes such as xanthine oxidase, cytochrome P450 (CYP450), and lipoxygenases. Metabolic reactions within peroxisomes and exposure to microbes, nanoparticles, and xenobiotics also contribute to ROS generation. Together, these diverse pathways highlight the multifactorial origins of ROS within biological systems, underscoring their central role in oxidative stress and cellular damage.

**Figure 2 medsci-13-00087-f002:**
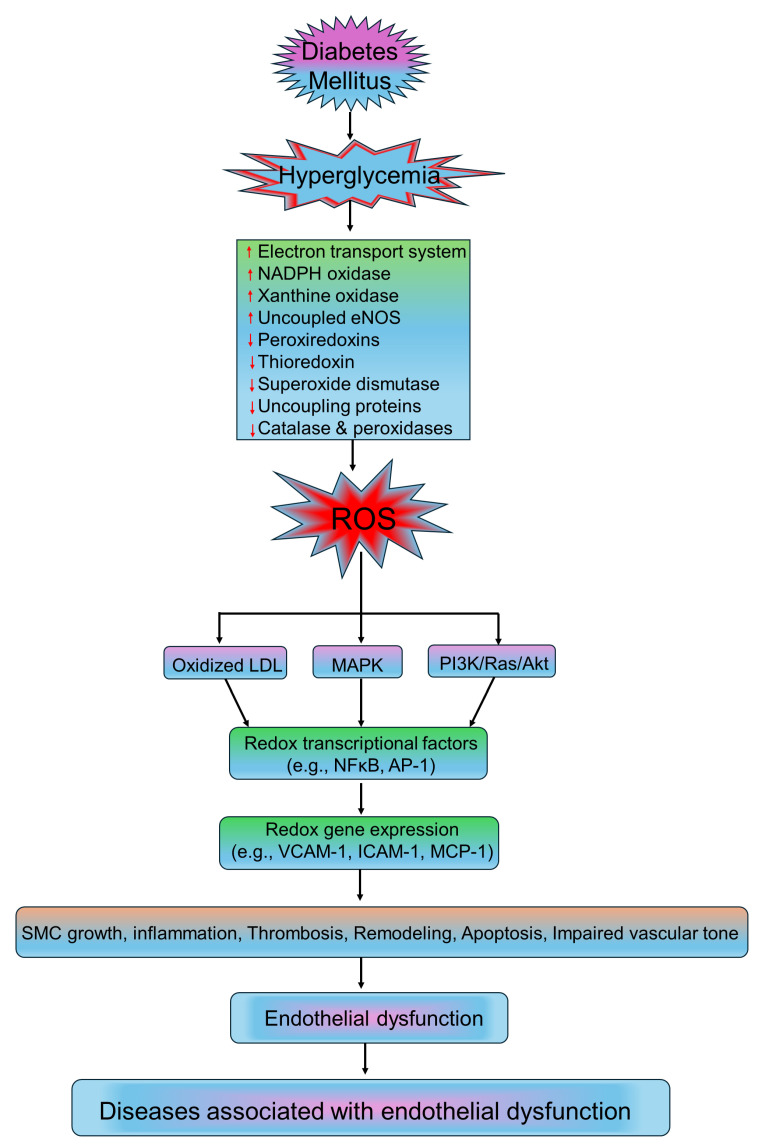
Mechanisms of oxidative-stress-induced endothelial dysfunction in DM. Hyperglycemia increases ROS production via upregulation of the electron transport system, NADPH oxidase, xanthine oxidase, and uncoupled eNOS, while decreasing antioxidant defenses such as peroxiredoxins, thioredoxin, superoxide dismutase, uncoupling proteins, and catalase/peroxidases. Elevated ROS activates signaling pathways including oxidized LDL, MAPK, and PI3K/Ras/Akt, which in turn stimulate redox-sensitive transcription factors (e.g., NFκB, AP-1), leading to increased redox gene expression of adhesion molecules and chemokines (e.g., VCAM-1, ICAM-1, MCP-1). These molecular changes promote smooth muscle cell (SMC) growth, inflammation, thrombosis, vascular remodeling, apoptosis, and impaired vascular tone, culminating in endothelial dysfunction and contributing to the development of diseases associated with endothelial dysfunction in diabetes mellitus. NADPH, Nicotinamide adenine dinucleotide phosphate; eNOS, endothelial nitric oxide synthase; ROS, reactive oxygen species; LDL, low-density lipoprotein; MAPK, mitogen-activated protein kinase; PI3K, phosphoinositide 3-kinase; Ras, rat sarcoma virus; Akt, protein kinase B; NFκB, nuclear factor kappa B; AP-1, activator protein 1; VCAM-1, vascular cell adhesion molecule 1; ICAM-1, intercellular adhesion molecule 1; MCP-1, monocyte chemoattractant protein 1; SMC, smooth muscle cell. The upward arrow indicates an increase, and the downward arrow indicates a decrease.

**Figure 3 medsci-13-00087-f003:**
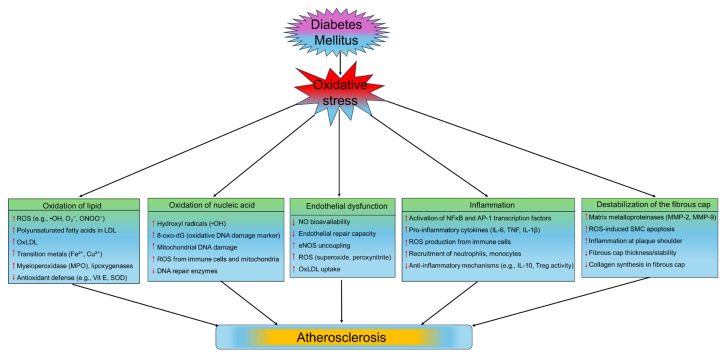
Pathophysiological mechanisms of oxidative stress-induced atherosclerosis. Oxidative stress leads to the oxidation of lipids, indicated by increased ROS and oxidized OxLDL, and the oxidation of nucleic acids, reflected by hydroxyl radicals, mitochondrial DNA damage, and oxidative DNA damage markers such as 8-oxo-dG. Endothelial dysfunction occurs due to decreased NO bioavailability, reduced endothelial repair capacity, and increased ROS from immune cells and mitochondria, leading to enhanced OxLDL uptake. Additionally, oxidative stress activates inflammation via NFκB and AP-1 transcription factors, increasing pro-inflammatory cytokines (IL-6, TNF, IL-1β), ROS from immune cells, and recruitment of neutrophils and monocytes, while influencing anti-inflammatory mechanisms (e.g., IL-10, Treg activity). Finally, oxidative stress destabilizes the fibrous cap through increased matrix metalloproteinases (MMP-2, MMP-9), ROS-induced smooth muscle cell apoptosis, and inflammation at the plaque shoulder, leading to reduced fibrous cap stability and collagen synthesis. Collectively, these processes contribute to the development and progression of atherosclerosis in the context of diabetes mellitus and oxidative stress. ROS, reactive oxygen species; •OH, hydroxyl radical; O_2^−^_, superoxide anion (or superoxide radical); ONOO^−^, Peroxynitrite anion; OxLDL, oxidized low-density lipoprotein; Vit E, vitamin E; SOD, Superoxide Dismutase; 8-oxo-dG, 8-oxo-2′-deoxyguanosine; NO, nitric oxide; eNOS, endothelial nitric oxide synthase; NFκB, nuclear factor kappa B; AP-1, activator protein 1; IL-6, interleukin 6; TNF, tumor necrosis factor; IL-1β, interleukin 1 beta; IL-10, interleukin 10; Treg, regulatory T cell; MMP-2, matrix metalloproteinases 2; MMP-9, matrix metalloproteinases 9; SMC, smooth muscle cell. The upward arrow indicates an increase, and the downward arrow indicates a decrease.

**Figure 4 medsci-13-00087-f004:**
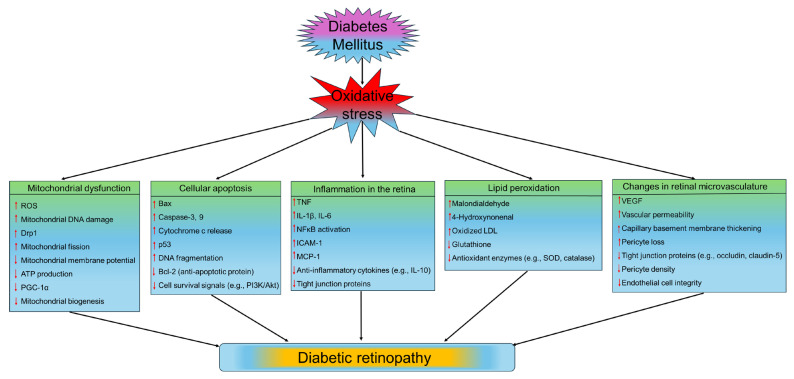
Pathophysiological mechanisms of oxidative stress-induced DR. Chronic hyperglycemia in diabetes induces oxidative stress, which leads to mitochondrial dysfunction characterized by increased ROS production, mitochondrial DNA damage, altered fission/fusion dynamics (e.g., Drp1), loss of membrane potential, impaired ATP production, and reduced mitochondrial biogenesis and PGC-1α activity. Oxidative stress also promotes cellular apoptosis in retinal cells through upregulation of pro-apoptotic markers such as Bax, caspases 3 and 9, cytochrome c release, p53 activation, and DNA fragmentation, while downregulating anti-apoptotic proteins like Bcl-2 and survival pathways such as PI3K/AKT. Concurrently, inflammation is amplified via elevated TNF-α, IL-1β, IL-6, ICAM-1, MCP-1, and NF-κB activation, along with altered expression of tight junction and anti-inflammatory proteins. Oxidative stress also accelerates lipid peroxidation, evidenced by increased malondialdehyde, 4-hydroxynonenal, and oxidized LDL, and reduced antioxidant defenses including glutathione and enzymes like SOD and catalase. These cumulative changes lead to structural and functional alterations in the retinal microvasculature, including increased VEGF expression, vascular permeability, basement membrane thickening, pericyte loss, compromised tight junctions (e.g., occludin, claudin-5), reduced pericyte density, and impaired endothelial integrity. Collectively, these molecular and cellular disturbances culminate in the onset and progression of diabetic retinopathy. ROS, reactive oxygen species; Drp1, dynamin-related protein 1; ATP, adenosine triphosphate; PGC-1α, peroxisome proliferator-activated receptor gamma coactivator 1-alpha; Bax, Bcl-2–associated X; DNA, deoxyribonucleic acid; Bcl-2, B-cell lymphoma 2; PI3K, Phosphoinositide 3-kinases; Akt, protein kinase B; TNF, tumor necrosis factor; IL-1β, Interleukin-1 beta; IL-6, interleukin 6; NFκB, nuclear factor kappa B; ICAM-1, intercellular adhesion molecule 1; MCP-1, monocyte chemoattractant protein-1; IL-10, interleukin 10; LDL, low-density lipoprotein; SOD, Superoxide dismutase; VEGF, vascular endothelial growth factor. The upward arrow indicates an increase, and the downward arrow indicates a decrease.

**Figure 5 medsci-13-00087-f005:**
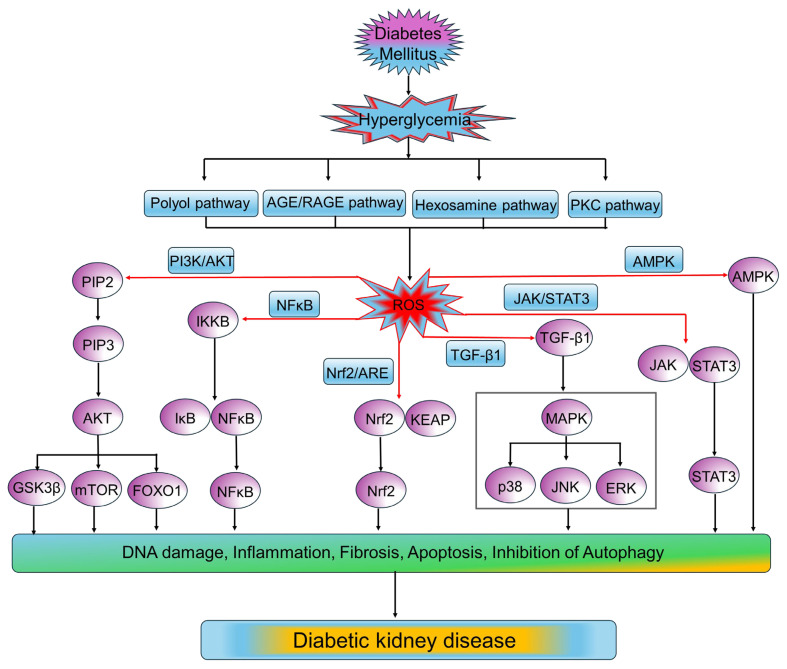
Pathophysiological mechanisms of oxidative stress-induced DKD. Hyperglycemia activates multiple metabolic pathways, including the polyol, AGE/RAGE, hexosamine, and PKC pathways—which collectively elevate ROS levels. ROS function as central mediators that disrupt key intracellular signaling cascades such as PI3K/AKT, NF-κB, Nrf2/ARE, JAK/STAT3, TGF-β1/MAPK, and AMPK pathways. These disruptions result in downstream effects including impaired antioxidant defense (via inhibition of Nrf2), enhanced inflammation (through NF-κB and JAK/STAT3), fibrotic signaling (via TGF-β1 and MAPK components p38, JNK, and ERK), and inhibition of autophagy (through suppressed AMPK and altered mTOR signaling). Collectively, these processes contribute to DNA damage, chronic inflammation, extracellular matrix deposition, apoptosis, and cellular dysfunction in renal tissue, ultimately culminating in the development and progression of diabetic kidney disease. AGE, advanced glycation end-products; RAGE, Receptor for AGE; PKC, protein kinase C; ROS, reactive oxygen species; PI3K, phosphoinositide 3-kinase; AKT, protein kinase B; AMPK, AMP-activated protein kinase; NFκB, nuclear factor kappa B; JAK, Janus kinase; STAT3, signal transducer and activator of transcription 3; TGF-β1, transforming growth factor-beta 1; Nrf2, nuclear factor erythroid 2–related factor 2; ARE, antioxidant response element; PIP2, phosphatidylinositol 4,5-bisphosphate; PIP3, phosphatidylinositol (3,4,5)-trisphosphate.

**Table 1 medsci-13-00087-t001:** List of factors that are altered in diabetes-induced oxidative stress.

Pathways	Altered Factors in Oxidative Stress	References
Polyol pathway	- ↑ Aldose reductase activity (glucose → sorbitol) - ↓ NADPH, impairing antioxidant defense - ↑ NADH, promoting ROS - ↑ Osmotic and oxidative stress from sorbitol accumulation	[165,166]
AGE/RAGE pathway	- ↑ Formation of AGEs - ↑ RAGE expression and activation - ↑ Inflammatory cytokines (TNF, IL-6) - ↑ ROS generation - ↓ Glyoxalase detoxification	[167,168,169]
PKC pathway	- ↑ DAG production - ↑ PKC isoform activation (PKC-β, PKC-δ) - PKC → NOX activation → ↑ ROS - Endothelial dysfunction, inflammation	[170,171,172]
Hexosamine pathway	- ↑ GFAT activity - ↑ UDP-GlcNAc production - ↑ Protein O-GlcNAcylation - Dysregulation of OGT/OGA balance - ↑ ER stress, mitochondrial dysfunction	[173,174,175]

Abbreviations: AGE, advanced glycation end product; RAGE, receptor for AGE; ROS, reactive oxygen species; TNF, tumor necrosis factor; IL-6, interleukin-6; DAG, diacylglycerol; PKC, protein kinase C; NADPH, nicotinamide adenine dinucleotide phosphate; NADH, nicotinamide adenine dinucleotide; GFAT, glutamine:fructose-6-phosphate amidotransferase; UDP-GlcNAc, uridine uiphosphate N-acetylglucosamine; O-GlcNAcylation, O-linked β-N-acetylglucosaminylation; OGT, O-GlcNAc transferase; OGA, O-GlcNAcase; ER, endoplasmic reticulum. The upward arrow indicates an increase, and the downward arrow indicates a decrease.

**Table 2 medsci-13-00087-t002:** Antioxidant therapy failure in RCTs.

Compound	Biomarker Effects	Clinical Outcomes	References
CoQ10	↑ FMD (+1.45%, *p* < 0.02) in meta-analysis of 12 RCTs	No reduction in cardiovascular events	[491]
MitoQ	↑ Brachial artery FMD by 42%, ↓ Ox-LDL and aortic stiffness in older adults	No clinical outcomes assessed	[489]
Omega-3	↓ BP (modest), ↓ triglycerides; favorable lipid profile	No consistent benefit for mortality or CV events	[492,493]
Melatonin	No improvement in FMD or oxidative stress markers in high-salt intake study	No vascular benefit observed	[487]
NAC	↓ MDA in hemodialysis patients (600 mg BID)	No BP reduction; no hard CV outcomes assessed	[488]
Resveratrol	No significant changes in BP or CV events in hypertensive patients across RCTs	Inconsistent or no benefit for CV endpoints	[494]

Abbreviations: CoQ10, coenzyme Q10; FMD, Flow-mediated dilation; RCTs, randomized clinical trials; MitoQ, mitoquinone mesylate; BP, blood pressure; NAC, N-Acetylcysteine; MDA, Malondialdehyde; CV, cardiovascular. The upward arrow indicates an increase, and the downward arrow indicates a decrease.

## Data Availability

Not applicable.

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
