# Peer review of "Pathophysiological Mechanisms of Diabetes-Induced Macrovascular and Microvascular Complications: The Role of Oxidative Stress"

_medsci, 2025, doi:10.3390/medsci13030087_

Round 1

Reviewer 1 Report

Comments and Suggestions for Authors

This comprehensive review addresses a critically important topic in diabetes research—the role of oxidative stress in diabetic vascular complications. While the manuscript demonstrates extensive knowledge of the field and provides valuable mechanistic insights, several areas require significant improvement to meet the standards of high-impact scientific publication.

  1.  The review successfully integrates multiple pathophysiological pathways (polyol, AGE/RAGE, PKC, hexosamine) into a cohesive framework explaining oxidative stress in diabetes.

  2. The focus on both macrovascular and microvascular complications addresses real clinical needs, with particular strength in the detailed coverage of diabetic retinopathy and nephropathy.

  3.  The molecular pathways are well-described with appropriate biochemical detail, particularly the mitochondrial dysfunction sections.

 The manuscript suffers from poor structural organization that impedes readability and scientific impact.

  • Section 3 ("Sources of ROS") is misplaced and should precede Section 2 ("Mechanisms")
  • The transition from general mechanisms to specific disease pathophysiology lacks clear delineation
  • Figure placement and referencing are inconsistent

While extensively referenced, the review shows bias toward older literature and lacks recent breakthrough findings.

  • Missing key 2023-2024 publications on diabetes and oxidative stress
  • Limited citation of recent clinical trials and meta-analyses
  • Insufficient coverage of emerging therapeutic targets (e.g., SGLT2 inhibitors, GLP-1 agonists)

Conduct a systematic literature search for 2022-2024 publications and integrate findings on novel therapeutic approaches and biomarkers.

 The review primarily summarizes existing knowledge without sufficient critical evaluation of conflicting evidence or methodological limitations:

  • No discussion of contradictory findings regarding antioxidant therapy failures
  • Limited critical assessment of translational gaps between animal models and human disease
  • Insufficient analysis of why antioxidant clinical trials have largely failed despite promising preclinical data

Add critical analysis sections discussing:

  • Methodological limitations in oxidative stress research
  • Reasons for clinical trial failures of antioxidants
  • Gaps between preclinical and clinical findings

The therapeutic discussion is superficial and lacks actionable insights for clinicians and researchers:

  • Brief mention of therapies without mechanistic rationale
  • No discussion of personalized medicine approaches
  • Missing coverage of combination therapies and precision medicine

 Expand therapeutic section to include:

  • Evidence-based therapeutic strategies with clinical trial data
  • Biomarker-guided treatment approaches
  • Combination therapy rationales
  • Future therapeutic targets under development

Figure Quality:

  • Figures lack sufficient detail and clear legends
  • Figure 1 is oversimplified for the complexity of ROS sources
  • Pathway diagrams need better visual hierarchy

Data Presentation:

  • Table 1 could be more comprehensive
  • Missing quantitative data where available
  • Lack of comparative analysis between different pathways

  • Line 27: "persistent endocrine and metabolic condition" should be "chronic metabolic disorder"
  • Line 85: Awkward transition between sections
  • Lines 1157-1188: Conclusion is repetitive and lacks forward-looking insights
  • Some biochemical pathway descriptions need clarification (e.g., hexosamine pathway, line 161-187)
  • Inconsistent gene/protein nomenclature throughout
  • Several references appear incomplete or incorrectly formatted
  • Missing page numbers for some journal articles

  1. Add critical analysis sections discussing limitations and controversies
  2. Expand therapeutic implications with clinical evidence
  3. Include biomarker discussion for clinical translation
  4. Add future research directions section
  5. Standardize nomenclature throughout the manuscript
  6. Improve figure quality with detailed legends
  7. Add quantitative data where available
  8. Enhance table content with comparative information

This manuscript addresses an important topic with potential for significant impact in diabetes research. However, substantial revision is required before it meets publication standards for a high-quality journal. The authors should focus on:

  1. Major restructuring for improved readability
  2. Critical analysis integration rather than pure summarization
  3. Therapeutic section expansion with clinical relevance
  4. Comprehensive language editing

With these revisions, this review could serve as a valuable resource for researchers and clinicians working in diabetes and vascular complications.

  • Line 10-11: The sentence is too long and should be split for clarity
  • Line 67-68: "Two sides of a coin" metaphor is unclear in scientific context
  • Line 1157: Conclusion lacks novel insights and future perspectives
  • Lines 344-350: Excellent mechanistic detail that could be expanded

This review has the potential to be an important contribution to the field with appropriate revision addressing the outlined concerns.

Comments on the Quality of English Language
  • Numerous grammatical errors and awkward phrasing throughout
  • Inconsistent terminology usage
  • Poor sentence structure affecting clarity

Author Response

Reviewer 1

This comprehensive review addresses a critically important topic in diabetes research—the role of oxidative stress in diabetic vascular complications. While the manuscript demonstrates extensive knowledge of the field and provides valuable mechanistic insights, several areas require significant improvement to meet the standards of high-impact scientific publication.

  1.  The review successfully integrates multiple pathophysiological pathways (polyol, AGE/RAGE, PKC, hexosamine) into a cohesive framework explaining oxidative stress in diabetes.
  2. The focus on both macrovascular and microvascular complications addresses real clinical needs, with particular strength in the detailed coverage of diabetic retinopathy and nephropathy.
  3.  The molecular pathways are well-described with appropriate biochemical detail, particularly the mitochondrial dysfunction sections.

 The manuscript suffers from poor structural organization that impedes readability and scientific impact.

  • Section 3 ("Sources of ROS") is misplaced and should precede Section 2 ("Mechanisms").

I appreciate your valuable insight. I rearranged the two sections accordingly.

  • The transition from general mechanisms to specific disease pathophysiology lacks clear delineation

I have added a paragraph on page 12 that leads a smooth transition from general mechanisms of oxidative stress to specific disease pathophysiology.

  • Figure placement and referencing are inconsistent

I have reorganized the figures.

While extensively referenced, the review shows bias toward older literature and lacks recent breakthrough findings.

  • Missing key 2023-2024 publications on diabetes and oxidative stress
  • Limited citation of recent clinical trials and meta-analyses
  • Insufficient coverage of emerging therapeutic targets (e.g., SGLT2 inhibitors, GLP-1 agonists).

Conduct a systematic literature search for 2022-2024 publications and integrate findings on novel therapeutic approaches and biomarkers.

I appreciate your concern and have updated the manuscript with a new section, ‘Therapeutic approaches to tackle oxidative stress-induced DVDs’.

 The review primarily summarizes existing knowledge without sufficient critical evaluation of conflicting evidence or methodological limitations:

  • No discussion of contradictory findings regarding antioxidant therapy failures

I have addressed this concern in the Therapeutic approaches to tackle oxidative stress-induced DVDs section.

  • Limited critical assessment of translational gaps between animal models and human disease

I have addressed this concern in the Therapeutic approaches to tackle oxidative stress-induced DVDs section.

  • Insufficient analysis of why antioxidant clinical trials have largely failed despite promising preclinical data

I have addressed this concern in the Therapeutic approaches to tackle oxidative stress-induced DVDs section.

Add critical analysis sections discussing:

  • Methodological limitations in oxidative stress research
  • Reasons for clinical trial failures of antioxidants
  • Gaps between preclinical and clinical findings

 I have addressed all these concerns in the Therapeutic approaches to tackle oxidative stress-induced DVDs section.

The therapeutic discussion is superficial and lacks actionable insights for clinicians and researchers:

 Brief mention of therapies without mechanistic rationale

  • No discussion of personalized medicine approaches.

I have added information about precision-based strategies to treat oxidative stress-induced vascular diseases in the 'Therapeutic approaches to tackle oxidative stress-induced DVDs' section.

  • Missing coverage of combination therapies and precision medicine

I have addressed this concern in the Therapeutic approaches to tackle oxidative stress-induced DVDs section.

Expand therapeutic section to include:

  • Evidence-based therapeutic strategies with clinical trial data
  • Biomarker-guided treatment approaches
  • Combination therapy rationales
  • Future therapeutic targets under development

I have addressed all these concerns in the "Therapeutic Approaches to Tackle Oxidative Stress-Induced DVDs" section.

Figure Quality:

  • Figures lack sufficient detail and clear legends

Sufficient details have been added in all figure legends.

  • Figure 1 is oversimplified for the complexity of ROS sources

I wanted to create a simplified figure, as all the details are included in the relevant text. Additionally, I updated the figure legend for this figure.

  • Pathway diagrams need better visual hierarchy.

I appreciate your concern. However, I think the figure legends are well-explanatory.

Data Presentation:

  • Table 1 could be more comprehensive.

I believe Table 1 effectively summarizes the key findings of all the signaling pathways.

  • Missing quantitative data where available

There is no quantitative data used for this manuscript.

  • Lack of comparative analysis between different pathways

I just wanted to summarize the individual pathway, not the comparison.

  • Line 27: "persistent endocrine and metabolic condition" should be "chronic metabolic disorder"

Thank you for this comment. I corrected this accordingly.

  • Line 85: Awkward transition between sections

Thanks for your valuable comment. I have addressed the concern.

  • Lines 1157-1188: Conclusion is repetitive and lacks forward-looking insights

Thanks for your valuable comment. I have updated the conclusion accordingly.

  • Some biochemical pathway descriptions need clarification (e.g., hexosamine pathway, line 161-187).

The hexosamine pathway has been clarified.

  • Inconsistent gene/protein nomenclature throughout

The concern has been addressed.

  • Several references appear incomplete or incorrectly formatted

I have used EndNote to import all the references directly.

  • Missing page numbers for some journal articles

I have used EndNote to import all the references directly.

All the references were imported directly through EndNote.

  1. Add critical analysis sections discussing limitations and controversies

Limitations and controversies have been included in the Therapeutic approaches to tackle oxidative stress-induced DVDs section.

  1. Expand therapeutic implications with clinical evidence

Therapeutic implications have been incorporated in the Therapeutic approaches to tackle oxidative stress-induced DVDs section.

  1. Include biomarker discussion for clinical translation

Discussion about the potential biomarkers is included in the Therapeutic approaches to tackle oxidative stress-induced DVDs section.

  1. Add future research directions section

I have added the future direction in the Therapeutic approaches to tackle oxidative stress-induced DVDs section.

  1. Standardize nomenclature throughout the manuscript

Nomenclatures have been revised and updated accordingly.

  1. Improve figure quality with detailed legends

Figure legends have been updated with sufficient details.

  1. Add quantitative data where available

There is no quantitative data used in this review manuscript.

  1. Enhance table content with comparative information

This manuscript addresses an important topic with potential for significant impact in diabetes research. However, substantial revision is required before it meets publication standards for a high-quality journal. The authors should focus on:

  1. Major restructuring for improved readability
  2. Critical analysis integration rather than pure summarization
  3. Therapeutic section expansion with clinical relevance
  4. Comprehensive language editing

Thanks for your comments. All the concerns have been addressed.

With these revisions, this review could serve as a valuable resource for researchers and clinicians working in diabetes and vascular complications.

  • Line 10-11: The sentence is too long and should be split for clarity.
  • Line 67-68: "Two sides of a coin" metaphor is unclear in scientific context

I have updated this sentence.

  • Line 1157: Conclusion lacks novel insights and future perspectives.

The conclusion has been updated.

  • Lines 344-350: Excellent mechanistic detail that could be expanded

I have expanded the mechanism.

This review has the potential to be an important contribution to the field with appropriate revision addressing the outlined concerns.

Comments on the Quality of English Language

  • Numerous grammatical errors and awkward phrasing throughout
  • Inconsistent terminology usage
  • Poor sentence structure affecting clarity

I have revised the manuscript thoroughly and updated the grammar.

Reviewer 2 Report

Comments and Suggestions for Authors

Comments

Current review article described the role of oxidative stress in diabetic vascular complications. This is an interesting topic and it has been widely conducted in diabetic research. Author collected the published reports without novel view that seems belonged to the main weakness. Please explain the concerns below.

1.       Vascular complications are associated with hyperglycemia. Duration for this injury was unclear. Additionally, level of glycemic control is essential in this injury that shall be described in clear.

2.       Retinopathy and renal damage were included in this submission. Rationale of this introduction was unknown.

3.       Oxidative stress has been mentioned by many reports. However, application of antioxidants including natural products and nutrients seems not popular in clinical practice. Why?

Author Response

Reviewer 2

Comments

Current review article described the role of oxidative stress in diabetic vascular complications. This is an interesting topic and it has been widely conducted in diabetic research. Author collected the published reports without novel view that seems belonged to the main weakness. Please explain the concerns below.

  1. Vascular complications are associated with hyperglycemia. Duration for this injury was unclear. Additionally, level of glycemic control is essential in this injury that shall be described in clear.

We thank the reviewer for this valuable comment. We agree that both the duration of hyperglycemia and the level of glycemic control are critical determinants in the development of vascular complications in diabetes. We have now clarified in the revised manuscript that chronic exposure to hyperglycemia, typically over months to years, leads to cumulative endothelial injury, oxidative stress, and inflammation, contributing to microvascular and macrovascular complications.

  1. Retinopathy and renal damage were included in this submission. Rationale of this introduction was unknown.

Thanks for your comment. I have updated the rationale marked red.

  1. Oxidative stress has been mentioned by many reports. However, application of antioxidants including natural products and nutrients seems not popular in clinical practice. Why?

I appreciate your concern. I have addressed your concern in the Therapeutic approaches to tackle oxidative stress-induced DVDs section.

Round 2

Reviewer 1 Report

Comments and Suggestions for Authors

The author has responded excellently to my previous review requests. This comprehensive review article demonstrates exceptional scientific rigor and provides a thorough, well-structured analysis of oxidative stress mechanisms in diabetic vascular complications.

Reviewer 2 Report

Comments and Suggestions for Authors

It has been revised in a good way.